# EVALUATING LANGUAGE MODELS' EVALUATIONS OF GAMES

**Katherine M. Collins**[1,2,3*]    **Cedegao E. Zhang**[2]    **Graham Todd**[4]    **Lance Ying**[1,5]
**Mauricio Barba da Costa**[2]    **Ryan Liu**[3]    **Prafull Sharma**[2]    **Adrian Weller**[1]
**Ionatan Kuperwajs**[3]    **Lionel Wong**[6]    **Joshua B. Tenenbaum**[2,†]    **Thomas L. Griffiths**[3,†]

[1]University of Cambridge, [2]MIT, [3]Princeton University, [4]NYU, [5]Harvard University, [6]Stanford University
[*]Corresponding author: katiemc@mit.edu    [†]Co-senior authors

## ABSTRACT

Reasoning is not just about solving problems—it is also about evaluating which problems are worth solving at all. Evaluations of artificial intelligence (AI) systems primarily focused on problem solving, historically by studying how models play games such as chess and Go. In this paper, we advocate for a new paradigm that assesses AI systems' *evaluation* of games. First, we introduce a formalism for evaluating such evaluations. We then leverage a large-scale dataset of over 100 novel board games and over 450 human judgments to compare evaluations produced by modern language and reasoning models against those of people and symbolic computational agents. We consider two kinds of evaluative queries: assessing the payoff (or fairness) and the funness of games. These queries span two dimensions relevant to the design of evaluations of AI evaluations: how complex a query is to compute and how difficult a query is to quantify. Our results show that reasoning models are generally more aligned to people in their evaluations of games than non-reasoning language models. However, we observe a non-monotonic relationship: as models get closer to game-theoretic optimal, their fit to human data weakens. We also observe more "jaggedness" across models for assessing funness, in line with the greater difficulty of quantifying this query. Across queries and games, reasoning models show highly variable and unpredictable resource usage when assessing queries, pointing to the importance of imbuing more resource-rational meta-reasoning in language and reasoning models.

## 1   INTRODUCTION

The ability to play games has long been used as a measure of assessing reasoning in artificial intelligence (AI) systems. From chess (Turing, 1950; Campbell et al., 2002; Newell et al., 1958) to Go (Silver et al., 2016) to poker (Brown and Sandholm, 2018) and now ARC-AGI (ARC Prize Foundation, 2025) and Pokémon (Anthropic, 2025; Karten et al., 2025), AI systems have consistently been evaluated on their ability to play games. The AI community is ever-expanding the set of games used in these assessments—even inventing new games (Ying et al., 2025; Verma et al., 2025)—to test the flexibility of AI systems' reasoning. However, these efforts offer a partial picture of the general reasoning capacity of AI systems. Reasoning is not just about playing games or solving problems, but also evaluating higher order aspects of the problems themselves, like *whether a game is worth playing in the first place* (see Figure 1a;  Wong et al. 2025; Griffiths 2020; Chu et al. 2023; Getzels 1987).

There are many ways to evaluate a game, and they are not all equally interesting. Determining whether a game is cooperative or competitive, for instance, is often relatively trivial: it does not require substantial compute and the query itself is unambiguous. In contrast, assessing the expected payoff of an arbitrary game is more interesting—it requires precise and complex computation (e.g., over likely game states). Formally assessing whether a game is likely to be "fun" adds a further layer of complexity, given the difficulty of determining how to quantify the answer to such a question which in turn, may also be difficult to compute . Yet a measure, like those studied in utilitarian ethics, may be hard to quantify. That is it might be hard to quantify what values to assign to different quantities

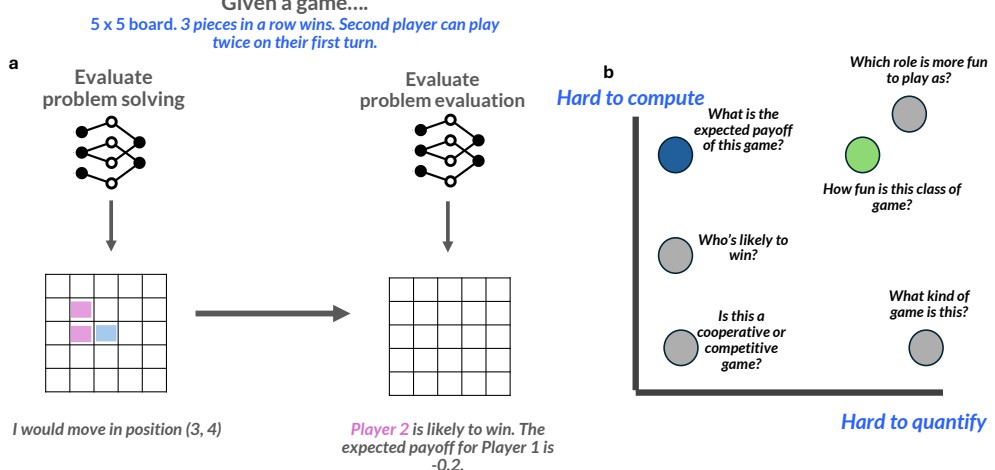

Figure 1: **Evaluating AI systems' evaluations**. **a,** A holistic understanding of model reasoning demands not just assessing how AI systems solve problems (play games), but how they evaluate whether problems, systems, or games are worth pursuing at all; **b,** Not all evaluations of problems are interesting for evaluating models. Good evaluation queries pose a challenge by being difficult to compute, difficult to quantify, or both.

(such as human lives or irreplaceable works of art), but it is easy to compute the sum of these values when considering a possible outcome of an action. This highlights **two dimensions of evaluations**: (1) difficulty to compute, and (2) difficulty to quantify (see Figure 1b). These dimensions are relevant when evaluating the evaluations produced by AI systems and inform the kind of human data we want to collect to compare these systems against. For example, human data may be more variable for queries that are harder to quantify (though also more relevant to real-world situations), and even a measure like payoff can still be difficult to quantify if it requires determining what counts as "reasonable" play over which to compute expected outcomes (see Appendix A7).

In this work, we lay out a **perspective on evaluating models on their capacity to—not just play—but evaluate games**. We then take initial steps to empirically assess language models on their capacity to conduct such evaluations. To do so, we draw on a corpus of 121 **novel games** from Collins et al. (2025). For each game, we test a series of language and reasoning models using two reasoning queries that engage both dimensions of evaluation: one that is difficult to compute and another that is difficult to compute and quantify. That is, we ask the models to evaluate: (1) **the expected outcome of the game** (from which we can compute the expected value or payoff), and (2) **the perceived funness of the game**. Critically, these evaluations are meant to capture reasoning about a game *before any actual play*, akin to someone deciding whether a game, goal, or task is worth their limited time and energy to engage with. We compare the evaluations produced by the models to those made by people and to a series of explicit gameplay (non-language-model) baselines. The baselines include both agents drawn from AI and computational cognitive modeling as well as, for games where it can be computed, the game-theoretic optimal payoff. Evaluating game evaluations **raises the question of what evaluations to measure against**—researchers may strive to build agentic systems that are as rational as possible (e.g., close to the game-theoretic optimal) or more human-aligned for effective thought partnering (Collins et al., 2024)—questions which may be even harder when an evaluation measure is difficult to quantify objectively (e.g., funness). We explore both directions in this work.

We find that non-reasoning language models, which directly produce game evaluations without using intermediate chains of thought, substantially differ from people's evaluations of games as well as the optimal game-theoretic expected payoff. These models' evaluations of the expected payoff of games are highly similar, even across different model families, suggesting that models may have picked up similar inductive biases about what makes a game fair based on shared training data. However, these evaluations remained substantially different from those produced by humans, indicating that such biases are insufficient for computing a human-aligned evaluation query. In

contrast, allowing models to reason through an intermediate chain-of-thought generally yields more sensible game evaluations relative to the game-theoretic optimal, but these are often still far from peoples' evaluations. Reasoning models are generally more aligned with both human judgments of expected payoffs and estimates obtained from our non-linguistic baselines. We observe a **non-monotonic relationship** between reasoning models and humans, where eventually an increase in alignment with the game-theoretic optimal solution begins to result in a decrease in alignment with human judgments. While reasoning models also generally capture human *funness* judgments better than non-reasoning language models, performance across models is inconsistent (e.g., more advanced models are not consistently more aligned to people in their funness evaluations), which matches the difficulty of quantifying "fun." And across both queries—we observe vastly different amounts of resources being used by reasoning models (as measured by reasoning tokens), motivating future work to **design more resource-rational problem-evaluation agents** capable of *dynamically adapting compute* to the evaluation query and problem at hand (Sui et al., 2025). We close with open questions that follow from a principled study of evaluating reasoners' abilities to evaluate.

## 2 FROM EVALUATING SOLUTIONS TO EVALUATING EVALUATION

One common way to study problem solving capacities is through games (see Section 5). A game $\mathcal{G}$ can be represented as a series of feasible states $\mathcal{S}$; possible actions $\mathcal{A}$; rules $\mathcal{T}$ specifying valid actions and state transitions given those actions; and one or more goal functions mapping from states $\mathcal{S}$ to possible rewards $\mathcal{R}$. Typically, problem solving (game play) is evaluated by assessing how well systems can estimate and deploy a policy $\pi_G(a_t \mid s_t)$ for choosing actions given a state to optimize reward $R_T$, where $T$ is the final turn or sum of the discounted reward over all timesteps (Sutton et al., 1998). The problem solving ability of an agent can also be assessed by measuring its efficiency in learning or estimating $\pi$ for the problem at hand.

However, real-world reasoning requires not just identifying what good actions are for any given problem or game state, but evaluating whether a problem or game is worth engaging with at all (Getzels, 1982; Nickles, 1981; Chu et al., 2023). For games, evaluation can be thought of as estimating some properties $\psi$ of a game $\mathcal{G}$. This may involve estimating $\pi$ as an intermediate step (see Collins et al. 2025), but critically places the emphasis of evaluation over the entire game rather than any single action and the reward of that action. Evaluating a game for a given query $\psi$ (e.g., whether a game is likely to be fun) may require breaking a query down into subqueries $\{\psi_1, \psi_2, ... \psi_f\}$ based on some factors $f \in \mathcal{F}$ from a space of factors (e.g., whether the game is fair; whether the game is likely to demand strategic thinking; how long the game is expected to be; etc.). Computing any $\psi_f$ may then also require varying levels of computation, as laid out in Figure 1b. Breaking down a larger query into different subqueries also raises the question of how solutions to these subqueries should be aggregated to answer the original question.

Evaluating a game itself inherently relies on less precise criteria than evaluating a player (for which victory or reward can be used). In addition, determining whether a problem is "good" might not permit objective evaluation.

This renders the task of problem evaluation more nebulous, yet accordingly, also more interesting. For instance, judgments to any query $\psi(G)$ could be compared to judgments made from other reasoners (e.g., people) or the judgments we may expect under a perfectly rational reasoner (e.g, game-theoretic optimal payoff, when it can be computed). Problem evaluators can also be assessed on the resource cost incurred, whether it is measured in wall-clock time, number of simulations run, or the number of reasoning tokens used.

## 3 METHODS

### 3.1 EVALUATIONS OVER NOVEL GAMES

We focus on the 121 two-player competitive strategy games playable on a grid from Collins et al. (2025). Games span a range of variants of Tic-Tac-Toe (see Appendix A2), most of which are novel in that they have not been publicly proposed before and therefore are both unlikely to have been played by people before and unlikely to be in the model' training data. While these games do represent a restricted space of the possible games one may play, many are strategically rich, capturing many

hallmarks of real decision making and planning problems people face. And already, this set already pushes productively away from the dominant focus in AI and psychology on one game at a time (e.g., Chess, Go, Diplomacy; see Appendix A1). Approximately 20 people evaluated each game per query (expected value and expected funness), totaling over 450 participants. People evaluated each game as "novices" *before* any actual play.

## 3.2 ELICITING MODEL GAME EVALUATIONS

We prompted a series of language and reasoning models to evaluate the expected payoff and funness of each of the 121 games (see Appendix A3.1). Models are sampled with 20 rollouts (to match the approximately 20 people who responded for each game query) using their default temperature (1.0 for o1, o3, and GPT-5, 0.7 for other models). In the main text, all reasoning model results are reported under medium reasoning effort; we explore other reasoning settings in Appendix A5.1. We also compare against a series of game reasoning models from Collins et al. (2025) which predict judgments by explicitly simulating gameplay between artificial agents. These agents vary in sophistication, ranging from random action selection, to a heuristic-based "Intuitive Gamer" model that approximates novice human gameplay, to models based on more extensive tree search, namely an "Expert" model that approximates depth-5 tree search based on van Opheusden et al. (2023), and a separate Monte Carlo Tree Search (MCTS, Coulom, 2006; Genesereth and Thielscher, 2014; Silver et al., 2016) based method (see Appendix A3.4).

## 3.3 EVALUATION MEASURES

Our primary measure of similarity is the $R^2$ between the averaged model judgments and human judgments (computed over all 121 games). We computed the split-half correlation between human participant judgments as a measure of the amount of explainable variance in the human data. Additionally, we compared models' and people's estimated payoffs in the subset of 78 games where we could compute an estimated game-theoretic optimal payoff (see Appendix A4.1). This allows us to also estimate the rationality of models relative to an estimated optimal payoff. We measure $R^2$, accuracy, and distance between the predicted and estimated optimal payoff. We also measure models' similarities to each other (within the same class of models, e.g., other reasoning or non-reasoning language models, or to other classes, e.g., a game reasoner that employs MCTS-based gameplay). We assess other measures of similarity in Appendix A4.

## 4 RESULTS

### 4.1 EVALUATING EXPECTED PAYOFF (FAIRNESS) OF GAMES

Non-reasoning language models are more similar to each other than they are to people's judgments or to tree-search based models, when comparing both the mean (Figure 2a) and distribution (Appendix Figure 8) over expected payoff of the games. This highlights some of the limits of inferring game properties purely from statistical associations in training data (i.e., without explicit reasoning or simulation). Non-reasoning language models which directly produce the game evaluation ("direct" prompting), without going through any intermediate chain-of-thought (CoT), tend to propose game evaluations that are even further from "optimally rational" (estimated game-theoretic) predicted payoffs (Table 1). Allowing non-reasoning models to produce a natural language chain-of-thought before coming to the final game evaluation yields both more rational and human-aligned fits to people, but still substantially less than more advanced reasoning models. These reasoning models are increasingly similar to both people (approaching the split-half human $R^2$ ($R^2 = 0.82$ [95% CI: 0.77, 0.86])) and the game-theoretic optimal (Table 1). However, we highlight a countervailing trend in the OpenAI family of models: initially, increasing sophistication (i.e., from GPT-4 to o1 and o3) corresponds with a better fit to both human judgments and game-theoretic judgments (Figure 1c; Figure 2a). But as sophistication continues to increase (i.e., from o3 to GPT-5), the fit to human judgments degrades even as the fit to game-theoretic judgments continues to improve (Figure 1c and Table 1), indicating worse alignment with semi-rational human participants (Figure 2b). Interestingly, however, a model like o3 can be instructed to approach game-theoretic optimal behavior, but it is harder to get GPT-5 to simulate novice human-like behavior (see Appendix A7).

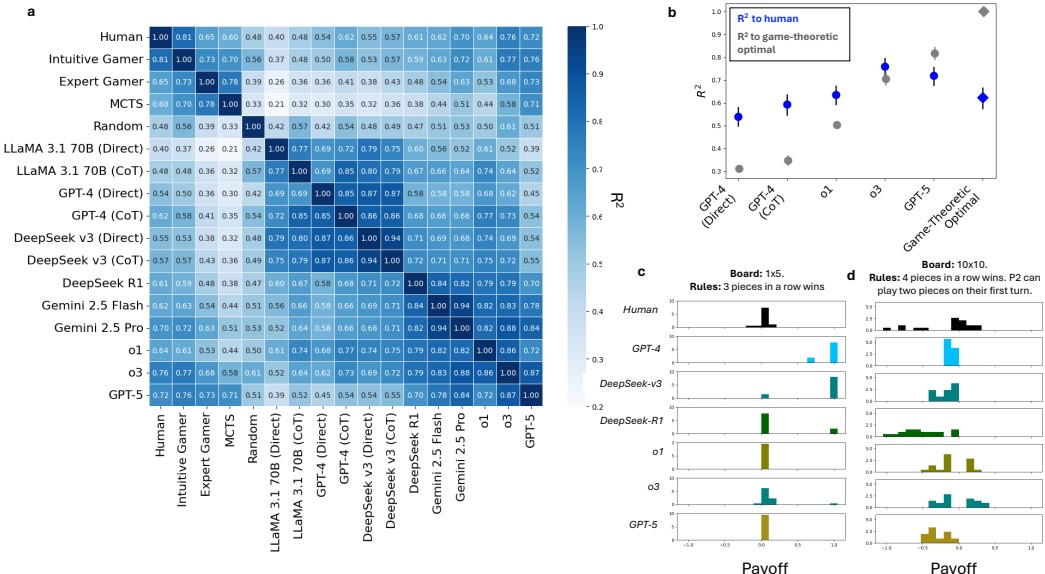

Figure 2: **Evaluating payoff (fairness) evaluations**. **a,** $R^2$ between human- and model-predicted payoff evaluations, over all 121 games. Each cell reports the $R^2$ in payoff evaluations between two reasoners. **b,** Payoff predictions across a subset of the OpenAI model family, compared to people's predicted payoffs (blue) and the estimated game-theoretic optimal (grey). Error bars depict bootstrapped $R^2$ 95% CIs. **c-d,** Example human- and a subset of model-predicted game evaluations. The distribution over human participants' judgments or each models' 20 rollouts are shown; the vertical axis shows the normalized density over binned distributions. Non-reasoning models (GPT-4 and DeepSeek-v3) are prompted with CoT. **c,** depicts a game where reasoning models are more aligned to people's evaluations; in other games as in **d,** judgments are highly varied across models—with no model faithfully capturing the rich structure in the distribution of human judgments. More example games are included in Appendix A4.

These results suggest that the kind of inductive biases picked up from standard pre- and post-training alone—at least on the kind of web and preference data these systems have been trained on—may be insufficient to model the kinds of judgments made by novice humans, even if they are able to capture some of the underlying game-theoretical dynamics. What then can allow models to move beyond the inductive biases baked in during standard training? Are reasoning models tasked with estimating game properties actually engaging in simulated play, such as sampling actions from some latent policy? We find that more advanced reasoning models are increasingly similar in both their aggregate and distributions of predicted judgments to game reasoners that use explicit simulation (see Figure 2a and Appendix Figure 8), but these fits are nuanced at a per-game level (see Figure 2c-d; Appendix A4.2). The close fit between the judgments of some reasoning models and search-based methods (e.g., o3 and the Intuitive Gamer model, or GPT-5 and MCTS) suggest that they could be engaging in some form of explicit simulation. While some closed source models do not expose the internal reasoning traces needed to progress on this question, we *can* inspect the reasoning traces of open reasoning models (e.g., DeepSeek R1). We find that while such models do engage in some kind of explicit game simulation in a portion of their games, the frequency of simulation is relatively low compared to other forms of reasoning (e.g., reasoning off of analogies, or even attempting to mathematically compute the expected payoff; see Appendix Table 4). More details on trace coding are included in Appendix A6. Moreover, even though reasoning models generally align better with people's predictions, there are still discrepancies at a per-game level (see Figure 2b-e and Appendix A4.2), underscoring the need for expanded analyses of language models' game evaluations. Future work can better understand how different evaluation strategies of models impact the distribution of their judgments relative to people, the "optimal" expected value, and other models' judgments.

| Reasoner | $R^2$ (95% CI) | Accuracy (95% CI) | Deviation (95% CI) |
|---|---|---|---|
| Human | 0.62 (0.58, 0.67) | 0.69 (0.65, 0.73) | 0.32 (0.31, 0.34) |
| Intuitive Gamer | 0.69 (0.66, 0.72) | 0.75 (0.72, 0.78) | 0.25 (0.24, 0.26) |
| Expert Gamer | 0.87 (0.85, 0.88) | 0.92 (0.91, 0.92) | 0.08 (0.08, 0.09) |
| MCTS | 0.89 (0.88, 0.91) | 0.91 (0.90, 0.92) | 0.06 (0.06, 0.07) |
| Random | 0.39 (0.34, 0.44) | 0.57 (0.55, 0.59) | 0.43 (0.41, 0.44) |
| LLaMA 3.1 70B (Direct) | 0.19 (0.17, 0.21) | 0.47 (0.45, 0.50) | 0.51 (0.50, 0.52) |
| GPT-4 (Direct) | 0.31 (0.30, 0.32) | 0.60 (0.59, 0.60) | 0.42 (0.41, 0.42) |
| DeepSeek v3 (Direct) | 0.35 (0.32, 0.38) | 0.61 (0.58, 0.64) | 0.42 (0.41, 0.43) |
| LLaMA 3.1 70B (CoT) | 0.30 (0.27, 0.33) | 0.48 (0.46, 0.50) | 0.48 (0.48, 0.49) |
| GPT-4 (CoT) | 0.38 (0.37, 0.39) | 0.59 (0.56, 0.60) | 0.42 (0.42, 0.43) |
| DeepSeek v3 (CoT) | 0.40 (0.37, 0.42) | 0.63 (0.59, 0.67) | 0.38 (0.37, 0.39) |
| DeepSeek R1 | 0.43 (0.37, 0.48) | 0.64 (0.59, 0.71) | 0.40 (0.38, 0.43) |
| Gemini 2.5 Flash | 0.53 (0.50, 0.55) | 0.79 (0.76, 0.82) | 0.30 (0.28, 0.31) |
| Gemini 2.5 Pro | 0.66 (0.64, 0.67) | 0.84 (0.82, 0.86) | 0.22 (0.21, 0.23) |
| o1 | 0.50 (0.49, 0.52) | 0.72 (0.69, 0.74) | 0.35 (0.34, 0.35) |
| o3 | 0.71 (0.68, 0.73) | 0.83 (0.81, 0.86) | 0.27 (0.26, 0.27) |
| GPT-5 | 0.82 (0.79, 0.84) | 0.88 (0.86, 0.90) | 0.15 (0.14, 0.16) |

Table 1: **Model and human predictions relative to the approximate game-theoretic optimal (GTO).** Human and model payoff evaluations are compared to the 78 of the 121 games where GTO payoff is estimable. Human and non-LLM results are reproduced from Collins et al. (2025). We compute their same metrics (accuracy, $R^2$, and absolute difference) to compare model predictions to GTO. $R^2$ is computed between a reasoner's average predicted payoff and approximate GTO for all games (higher is closer to GTO). Accuracy measures whether the predicted payoff is within 0.5 of the GTO payoff; for this subset of games, GTO payoff is always $\in \{-1, 0, 1\}$ (higher means closer to GTO). Deviation computes the average absolute difference between the predicted payoff and approximate GTO (lower means closer to GTO value). We report 95% bootstrap confidence intervals (CIs) in parentheses. For the computational models, CIs were computed by bootstrapped subsampling over simulated runs per game.

| Model | Balance | Challenge | Length | Strategic Richness | Novelty |
|---|---|---|---|---|---|
| LLaMA 3.1 70B (CoT) | 47.5% | 97.1% | 53.0% | 99.5% | 56.8% |
| GPT-4 (CoT) | 55.6% | 98.6% | 67.9% | 98.6% | 54.5% |
| DeepSeek v3 (CoT) | 71.7% | 95.7% | 70.9% | 98.1% | 65.4% |
| DeepSeek R1 | 85.7% | 99.7% | 90.8% | 99.3% | 62.3% |
| Gemini 2.5 Flash | 74.6% | 99.9% | 76.5% | 100.0% | 48.3% |
| Gemini 2.5 Pro | 86.2% | 100.0% | 77.5% | 100.0% | 73.8% |

Table 2: **Funness factors discussed in chain-of-thought and reasoning traces during evaluation.** Chain-of-thought and reasoning traces for models evaluating funness were inspected and automatically coded by o3 based on whether they mentioned particular factors (e.g., game balance, length, etc.) in their explicated evaluative process. Percent of reasoning traces mentioning each factor are averaged over each responses per game.

## 4.2 EVALUATING GAME FUNNESS

Next, we evaluate language models' judgments of the funness of games. Participants and models were instructed to define funness however they wished—this is by design: we are interested in model and people's assessments of how to even define fun in the first place (a query which is both "hard to quantify" and "hard to compute.") When models answered this without going through any intermediate chain-of-thought or reasoning trace, they consistently produced results that poorly matched people's judgments (Figure 3a). On the other hand, models that engage in some kind

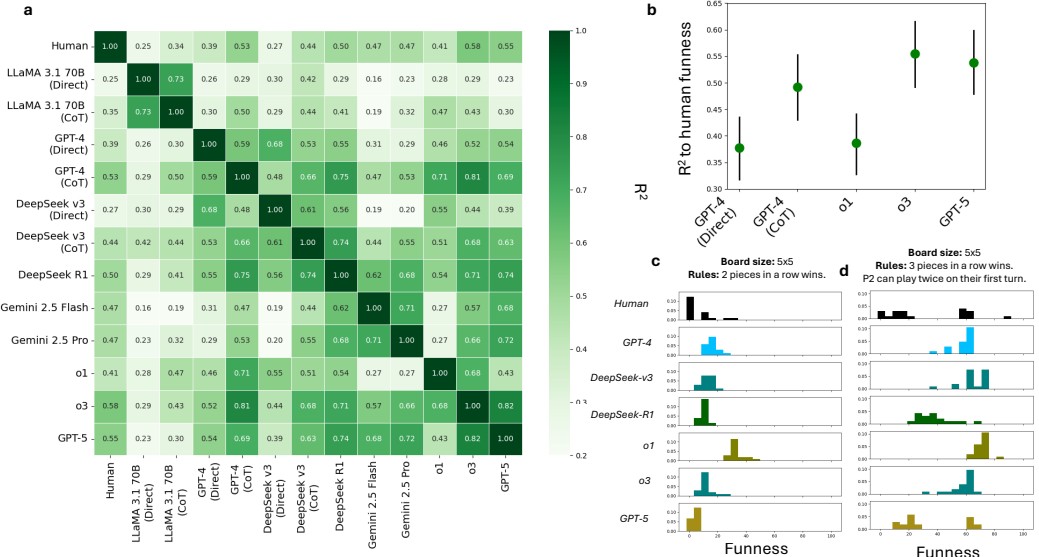

Figure 3: **Evaluating funness evaluations**. **a,** $R^2$ between human- and model-predicted funness evaluations, over all 121 games. Each cell reports the $R^2$ in funness evaluations between those two reasoners. **b,** Funness evaluations across a subset of the OpenAI model family reveals non-monotonicty in fits when moving from non-reasoning to reasoning models. Bootstrapped $R^2$ are computed relative to people's predicted funness, with error bars depicting the bootstrapped 95% CIs. **c-d** Example human- and model-predicted distributions over funness. The vertical axis shows the normalized density over the histogram of people and models' binned judgments. **c,** depicts an example where people and most models (though not all, e.g., o1) recognize that the game is unlikely to be fun; **d,** however, people's funness judgments are also variable, e.g., showing bimodality. This bimodality is not captured by most models, with a few exceptions (e.g., GPT-5) for this game. More example game evaluations are in Appendix A4.

of natural-language-based intermediate reasoning capture more of the variance in human data (see Appendix Figure 8). However, comparisons across model sophistication are more "jagged" (Karpathy, 2024)—both in capturing aggregate human judgments and the distribution of individuals' judgments (Figure 3a-b)—compared to our results measuring models' evaluations of expected payoffs of games. This is likely because evaluating funness is hard to quantify: estimating funness requires first determining what factors ($f \in \mathcal{F}$) make a game fun, then measuring the game along each of those metrics ($\psi_f$), and finally aggregating across those metrics in a sensible way. While we find that most of the chain-of-thought and reasoning trace-based models that we can examine consider the challengingness and strategic richness of games when assessing funness (Table 2), models differ in their rates of assessing game balance and length, potentially driving some of the disparities in the eventual scalar funness judgments (Figure 7).. Thus, these differences may arise from either disparities in how the values of each subquery are computed (which we may expect based on our evaluations of models' differences in evaluating expected game payoffs), or different methods in which these values are aggregated. We again identify differences in rates of explicit simulation of gameplay in some of the reasoning models (see Appendix Figure 16).

### 4.3 RESOURCE USAGE WHEN EVALUATING GAMES

How much compute are models engaging in to evaluate these games, and what may explain why some games and queries demand more compute? To begin to assess resource usage, we conduct an exploratory analysis into the number of reasoning tokens used by a series of reasoning models (DeepSeek-R1, Gemini 2.5 Flash and Pro, o3, and GPT-5) when determining game evaluations.

While there is some relationship between the number of tokens used when estimating game payoff across models (with the exception of DeepSeek-R1) there is minimal relation across models' token

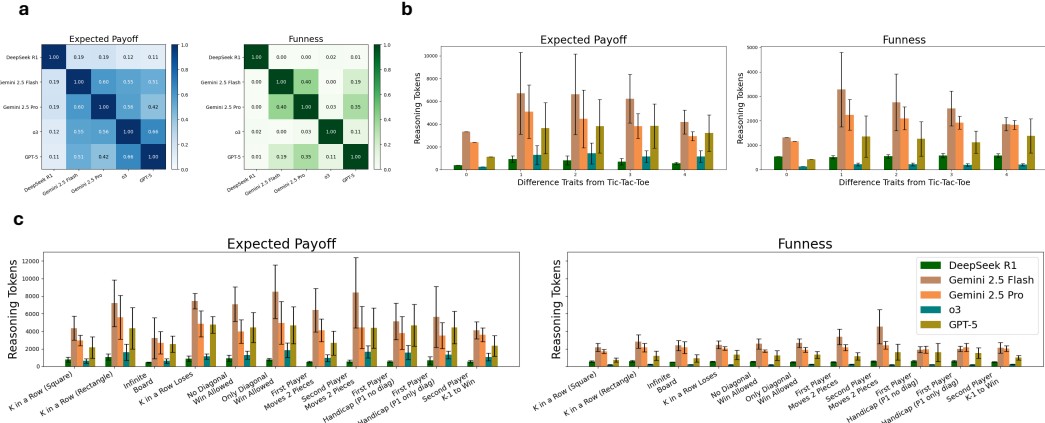

Figure 4: **Reasoning tokens used across games and evaluation queries. a,** $R^2$ between models' median number of reasoning tokens used per game, for the payoff and funness evaluation queries. **b,** Median reasoning tokens used for games based on how many "traits" they differ from Tic-Tac-Toe (e.g., a game that is not played on a $3 \times 3$ board, requires 4 pieces in a row to win, and constrains the win conditions to "only diagonals count," has 3 traits different from Tic-Tac-Toe). Tic-Tac-Toe itself is zero. The heights of bars show averaged number of median tokens for that game, with error bars depicting standard deviation over games. **c,** Token usage based on higher-level game category.

use when evaluating game funness (Figure 4a). There are also vast differences in the magnitude of tokens used across models and query type: despite funness being more ambiguous (and possibly involving multiple sub-questions to compute), models generally use far fewer tokens to estimate funness (Figure 4b-c).

Moreover, at a per-game level, while we may expect less typical games (e.g., games more distant from Tic-Tac-Toe) to demand more compute (reasoning tokens) to reason about, we did not observe a measurable difference between game "novelty" and token usage (Figure 4b-c), where "novelty" is measured as the number of features of a game that differ from the base Tic-Tac-Toe (e.g., if the game is not played on a $3 \times 3$ board, or involves asymmetric win conditions between players), as used in Collins et al. (2025). We also do not observe a strong relationship between token usage and the distance between the model's output and the game-theoretic optimal or human predictions (see Appendix Figure 15). This raises a question about what determines the expenditure of reasoning tokens and how models could be made more resource-rational in dynamically adapting compute (De Sabbata et al., 2024; Sui et al., 2025) based on game complexity or other factors (e.g., how precise an evaluation needs to be). These adjustments should likely be made under the particular resource limitations of language models, which may be different from those of people (Griffiths et al., 2015; Lieder and Griffiths, 2020). Additional examples and trace analyses are included in Appendix A6.1.1 and A6.3.

## 5 RELATED WORK

**Problem evaluation and metacognition in cognitive science** The meta-level problem of deciding which problem to solve is an active area of research in cognitive science to which our work directly relates. While people have remarkable cognitive flexibility to represent and reason about a wide range of problems—even posing new questions and new goals (Schulz, 2012; Chu et al., 2023; Getzels, 1982)—meta-reasoning is necessary because people have limited cognitive resources (Griffiths, 2020). Thus, resource-rational analysis (Icard, 2023; Lieder et al., 2025) has been especially successful as a framework for the development of computational models of problem selection in contexts such as problem representation and decomposition (Ho et al., 2022; Correa et al., 2023; Binder et al., 2023) and strategy selection (Lieder and Griffiths, 2017; Binz et al., 2022). Algorithms for human problem selection extend to various other domains as well, including deciding how much to plan given a set of alternatives (Sezener et al., 2019; Callaway et al., 2022; Kuperwajs et al., 2024) or when to

even engage with a task at all as opposed to quitting (Kuperwajs and Ma, 2022; Sukhov et al., 2023). Building AI systems that collaborate and interact with people requires understanding not just how machines and people solve problems, but also how to evaluate novel problems from the perspectives of both AI systems and humans.

**Assessing language model reasoning**   Prior work has predominantly investigated the reasoning capabilities of language models with the goal of solving problems instead of evaluating them. These broad efforts span topics such as math and symbolic reasoning (e.g., Mirzadeh et al., 2025; Sprague et al., 2025; Holliday et al., 2024), coding (e.g., Yang et al., 2025), psychology and behavioral economics tasks (e.g., Liu et al., 2025b; Piedrahita et al., 2025), vision/multimodal tasks (e.g., Chen et al., 2024; Zhang et al., 2024), planning and robotics (e.g., Kambhampati et al., 2024; Wang et al., 2025), linguistic phenomena (e.g., Hu et al., 2022; Qiu et al., 2025), and games (see additional related work, Appendix A1). These works typically evaluate reasoning models (e.g., OpenAI, 2025) or prompt-induced reasoning such as chain-of-thought (Wei et al., 2022; Nye et al., 2021; Kojima et al., 2022) against non-reasoning baselines. A general finding across these studies is that newer and larger models enable better reasoning capabilities (Mirzadeh et al., 2025)—sometimes with the help of tools such as program executors or additional training techniques (Yang et al., 2025), and such interventions may make reasoning more efficient (De Sabbata et al., 2024; Sui et al., 2025; Liu et al., 2025b; Zhang et al., 2025). Studies have also found that reasoning models' reasoning token usage may co-vary with human reaction times across several tasks (de Varda et al., 2025). We additionally contrast our work with LLM-as-judge approaches in Appendix A1.

**Applying methods from cognitive science to understand language models**   Our work follows a well-established line of recent research that employs psychological findings to better understand language model behavior (e.g., McCoy et al., 2024a;b; Binz and Schulz, 2023; Ku et al., 2025; Coda-Forno et al., 2024; Frank, 2023). Such research typically replicates an existing psychological study by replacing participants with language models, which are compared the original participants as well as rational cognitive models that describe desired behavior (e.g., Liu et al., 2024; Marjieh et al., 2024; Liu et al., 2025a; Zhu and Griffiths, 2024). Insights from cognitive science have also contributed to designing and improving LLM-based agents (Liu et al., 2026). Additional related work is in Appendix A1.

## 6 DISCUSSION

A holistic understanding of AI systems' reasoning capacities requires understanding not only how models solve problems, but also how they assess problems. Games are a microcosm of the kind of systems of rules and rewards that we want to use AI to evaluate. We show that while language can capture a substantial amount of associative knowledge that can be brought to evaluate new systems (e.g., whether a game is fun), language-based intuitions alone without some deliberative, iterative thinking can only go so far. For these games and queries, some form of simulation or explicit reasoning seems essential for aligning with human judgments and computing the optimal game-theoretic value. Our work opens up a range of questions for future work, namely:

- Whose evaluations should models be evaluated against: people (of which group and what level of experience), or the "optimally rational" evaluation?

- What cost are we willing to pay for such evaluations? How should we balance resource demands to evaluate problems (particularly when deciding whether to engage more with, or even solve, the problem in the first place)?

- What inductive biases have models already picked up about what makes a problem or game fair, or worth engaging with? What are the limits about what can be learned from generic supervised training vs. reinforcement learning?

- Are language and/or reasoning models interally engaging in some kind of game simulation in order to produce tokens, beyond what is written explicitly in chain-of-thought rationales or reasoning traces?

- How can models be encouraged to explicitly simulate when evaluating? When should they, and when should they not simulate—given a particular compute budget?

- What other evaluation queries should we engage with, to understand models' capacities for more general problem evaluation and meta-reasoning?

Indeed, our work only scratches the surface of evaluations of game evaluation. It is an open empirical question how far our results generalize to other competitive board games (like Hex or Othello variants) and entirely different categories of games (e.g., cooperative games, or games with asymmetric roles) and other settings (e.g., in law and finance) which may require asking other evaluation queries and designing new human experiments to compare models against. Our assessments are not meant to be definitive: model performance is sensitive to a host of factors like the exact prompt (here, we prompt participants with the human instruction text; see Appendix A3.1 and Appendix A7) and other hyperparameters (e.g., reasoning amount; see Appendix A5.1). Future work should better explore the relationship between the style of player or agent a model is simulating when making such assessments (see Appendix A7). We also note that our evaluations focus on "novice" game reasoners; it is an open question of how well models relate to people of varying skills, culture, and experience. One of the goals of this work is to carve out an underexplored space of evaluation of AI systems: evaluating their capacity to evaluate problems; our empirical work only scratches the surface of this rich space.

Evaluating AI systems' evaluations is important for building human-beneficial AI thought partners (Collins et al., 2024) that meet our expectations for deciding what problems to solve (e.g., in educational contexts) or determining whether a system is fair. The latter is especially important if AI systems are used in part to create new rules that people engage with or are guided by (Koster et al., 2022; Tacchetti et al., 2025). For example, it is important that AI systems involved in automated mechanism design (Myerson, 1983; Maskin, 2008; Hurwicz, 1973; Milgrom, 2004) can appropriately evaluate whether the resulting system will be fair (and even engaging) for other people to participate in. Moreover, studying where models differ from people in their evaluations of systems can also inform the construction of other kinds of thought partners that complement people (e.g., as cognitive prostheses; Lieder et al., 2019) by adjusting people's expectations about a new problem or system. We hope our work paves the way for future evaluations—evaluations that go beyond assessing model problem solving, but flexible problem and system evaluation.

## 7    CONCLUSION

Reasoning is not just about solving problems, but evaluating whether problems are worth solving at all. We laid out a perspective on thinking about the evaluation of AI systems' capacity for problem evaluation. We focused on the domain of games, particularly, two-player competitive strategy board games and assessed a series of language and reasoning models on their judgments about games over two evaluation queries—payoff and funness—that span a range of difficulty to compute and to quantify. Our work raises questions for how to design more human-beneficial and resource-efficient machine reasoners and evaluate their evaluations of whether new problems are worth solving.

## ETHICS STATEMENT

Our work is directly related to AI alignment, here assessing whether AI aligns with human judgments of fairness and funness of games. This novel perspective—understanding whether models come to similar conclusions about what makes a system, or game, fair—has broad implications for AI models that are designing systems of rules that may involve or engage people. Our work also looks at how models' judgments of funness compare to people; models which better anticipate what people will find engaging could be used to optimize the design of highly engaging games, which could cross the threshold toward being (harmfully) addictive. The use of more human-aligned models of human engagement is not all fun and games, warranting careful consideration of how it is used in practice.

## ACKNOWLEDGMENTS

We thank Simon Frieder, Shubhra Mishra, Gabriel Poesia, Sam Cheyette, and Tracey Mills for valuable conversation that informed this work. KMC acknowledges support from the Cambridge Trust and King's College Cambridge. This work was supported in part by AFOSR (FA9550-22-1-0387), the ONR Science of AI program (N00014-23-1-2355), a Schmidt AI2050 Fellowship to JBT,

and the Siegel Family Quest for Intelligence at MIT. TLG acknowledges support from the NOMIS Foundation. AW acknowledges support from a Turing AI Fellowship under grant EP/V025279/1, The Alan Turing Institute, and the Leverhulme Trust via CFI. IK acknowledges support from the National Science Foundation (IIS-2312373).

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

APPENDIX

## A1   ADDITIONAL RELATED WORK

**Games and the evaluation of AI**   Our work is related to a line of research that uses games to benchmark and understand AI model's capabilities. Games have long served as valuable environments for evaluating AI models and algorithms (Shannon, 1950; Newell, 1955; Campbell et al., 2002; Mnih et al., 2015; Silver et al., 2016; Yannakakis and Togelius, 2018; Vinyals et al., 2019; Bard et al., 2020; FAIR et al., 2022; van Opheusden et al., 2023; Bailis et al., 2024; Todd et al., 2025; Shojaee et al., 2025). Games are useful for evaluation in part because they offer precise rules and reward structures that are easily encoded into artificial systems while still requiring players to engage in a variety of complex cognitive behaviors, from long-range planning to semantic understanding to social inference—as has made games especially appealing for studying LLM decision making (Wu et al., 2024; Duan et al., 2024; Huang et al., 2025; Lin et al., 2025). However, rather than study LLM play in games, here, we focus on studying how LLMs *evaluate* games, before any play, and compare such evaluations to humans' evaluations. Additionally, our focus on game variants that are unlikely to have been previously studied and are unlikely to be present in extant training corpora aligns with a

recent trend to focus on novel or generated games for the purpose of evaluating modern AI systems (Ying et al., 2025; Verma et al., 2025).

**Language models as judges** Lastly, one parallel application in which language models are also used as evaluators is in LLM-as-a-judge paradigms (Li et al., 2024). In these settings, LLMs are used to provide evaluations by leveraging their ability to process diverse data types and provide scalable assessments that approximate human preferences (Zheng et al., 2023). Such methods have been applied for generating various scores (e.g., Bai et al., 2023), answering yes/no questions (e.g., Shinn et al., 2023), and conducting pairwise comparisons (e.g., Liu et al., 2025c), which have been used to improve aspects of models (e.g., Dubois et al., 2023), data (e.g., Zhang et al., 2023), agents (e.g., Zhuge et al., 2025), and even reasoning (Lightman et al., 2023). However, unlike this literature—or other literature evaluating the kinds of evaluations used to test AI systems, e.g, (Zhou et al., 2025)— our motivation is not to use language model judgments to acquire assessments at scale. Instead, our work focuses on the cognitive traits of these models—using the setting of games to analyze how language models compare to humans in reasoning about tasks that are difficult to compute or quantify.

## A2   EXAMPLE GAMES

The novel games we explore here span a wide range of board sizes and shapes, as well as game rules. We provide several example games, broken down by game categories in Table 3 below.

| Game Category | Example Game |
|---|---|
| K in a Row (Square) | 7 pieces in a row wins on a $10 \times 10$ board |
| K in a Row (Rectangle) | 4 pieces in a row wins on a $4 \times 9$ board |
| Infinite Board | 5 pieces in a row wins on an infinite board |
| K in a Row Loses | A player loses if they make 3 pieces in a row on a $4 \times 4$ board |
| No Diagonal Win Allowed | 4 pieces in a row wins on a $10 \times 10$ board, but a player cannot win by making a diagonal row |
| Only Diagonal Win Allowed | 4 pieces in a row wins on a $5 \times 5$ board, but a player can only win by making a diagonal row |
| First Player Moves 2 pieces | 3 pieces in a row wins on a $3 \times 3$ board; Player 1 can place 2 pieces as their first move |
| Second Player Moves 2 Pieces | 10 pieces in a row wins on a $10 \times 10$ board; Player 2 can place 2 pieces as their first move |
| First Player Handicap (P1 no diag) | 3 pieces in a row wins on a $3 \times 3$ board, but Player 1 cannot win by making a diagonal row |
| First Player Handicap (P1 only diag) | 4 pieces in a row wins on a $7 \times 7$ board, but Player 1 can only win by making a diagonal row |
| Second Player K-1 to Win | Player 1 needs 3 pieces in a row, but Player 2 only needs 2 pieces to win on a $5 \times 5$ board |

Table 3: **Game categories and example games.** The 121 games can be grouped into categories based on their board shape and game rules. Example games are shown for each category.

## A3   ADDITIONAL MODEL DETAILS

### A3.1   PROMPTS AND ADDITIONAL LANGUAGE MODEL GENERATION DETAILS

Models were prompted with a lightly-modified version of the human instruction text from (Collins et al., 2025). Experiment instructions were provided in the "system" prompt, with the specific game provided in the "user" prompt. For payoff questions, models were prompted (like people) to provide separate estimates $P(\text{P1 wins}|\text{not draw})$ and $P(\text{ends in a draw})$. Responses were provided simultaneously. These scores were combined into a single measure of payoff, i.e., $P(\text{P1 wins}) = P(\text{P1 wins}|\text{not draw}) \times (1 - P(\text{ends in a draw}))$ and payoff for Player 1 is $\left(1 - (P(\text{ends in a draw}) + P(\text{P1 wins}))\right) \cdot (-1) + P(\text{P1 wins})$. Future work can explore eliciting payoff directly in a single query. Models were asked (again, like people) to estimate the funness of the game, with respect to the broader class of games.

For non-thinking models, we varied whether they were prompted to respond directly (just a number) or via "chain-of-thought" (CoT) (Wei et al., 2022). Further details are provided when describing our task prompt. Any run for a language model that was prompted to directly answer the question (i.e., without going through a CoT first) and still outputed a natural language rational first was filtered out.

Thinking models were all prompted in CoT fashion, with the exception of DeepSeek-R1 which required a few modifications: for R1 specifically, we append the system prompt in the primary "user" prompt, per recommendations on Together AI API. We additionally adjusted the maximum tokens to $32,000$ tokens as we observed that R1 tended to respond longer than the default. Any run that took over the limit was filtered out.

## A3.2  System prompt

---

**System prompt for payoff evaluation**

```
Welcome! We are conducting an experiment to understand how people think about games.
Your answers will be used to inform cognitive science and AI research.

In this experiment, you will be reading short descriptions of board games and answering
two simple questions about each game.

Each game is played by players who take turns by placing pieces on a grid, similar to
games like Connect 4, Gomoku (5-in-a-row), or Tic-Tac-Toe.

You will be reading descriptions of games in which the size of the board and the rules
for winning vary. We will always show you an example game board from each description.
For example, you might read a description like:
- The board in this game is a 5x5 grid.
- In this game, the rule is that the first player to make 3 in a row wins.

Then, for each game, your task is to answer: assuming both players play reasonably -- if
the game does not end in a draw, how likely is it that the first player is going to win
(not draw), and how likely is a draw

You will answer this question by providing a response (in the form of a number) between
0 and 100.

Before you answer the question for each game, you will have as much time as you want to
think about the game and its rules.

Afer you feel like you understand the game, you can provide your response.

For each game, you can write on a scratchpad to think about the game before you answer.

We encourage you to take your time and carefully analyze the game before providing your
answer.
```

---

**System prompt for funness evaluation**

```
 Welcome! We are conducting an experiment to understand how people think about games.
Your answers will be used to inform cognitive science and AI research.

In this experiment, you will be reading short descriptions of board games and answering
a simple question about each game.

Each game is played by players who take turns by placing pieces on a grid, similar to
games like Connect 4, Gomoku (5-in-a-row), or Tic-Tac-Toe.

You will be reading descriptions of games in which the size of the board and the rules
for winning vary. We will always show you an example game board from each description.
For example, you might read a description like:
- The board in this game is a 5x5 grid.
- In this game, the rule is that the first player to make 3 in a row wins.",

Then, for each game, your task is to answer: how fun the game is to play

You will answer this question by providing a response (in the form of a number) between
0 and 100.

We ask that you think about funness with respect to this kind of game; that is, games
that involve players placing pieces on a grid. You can define fun however you wish.

Before you answer the question for each game, you will have as much time as you want to
think about the game and its rules.
```

```
Afer you feel like you understand the game, you can provide your response.

For each game, you can write on a scratchpad to think about the game before you answer.

We encourage you to take your time and carefully analyze the game before providing your
answer.
```

## A3.3   TASK PROMPT

Below are two example task prompts (specified in the "user" part of the prompt). Note that "You may first write out your thoughts on a scratchpad." is included for the "CoT" variant (and removed for the "Direct" variant). As noted, we filter out any run in the "Direct" variant that includes a "chain-of-thought" response before providing a number (for the LLaMA 3.1 70B, GPT-4, and DeepSeek v3 "Direct" variants).

**Example payoff evaluation prompt, for an example game**

```
Imagine you are playing the following game:

Board size: 3 x 5
Win conditions: 3 pieces in a row wins.

You will answer two questions. For each question, provide your a single number between 0
and 100.

Q1:
If the game does not end in a draw, assuming both players play reasonably, how likely is
it that the first player is going to win (not draw)?

Answer on a scale of 0 to 100.
Let 0 = "First player definitely going to lose",
50 = "Equally likely to win or lose",
100 = "First player definitely going to win"

Q2:
Assuming both players play reasonably, how likely is the game to end in a draw?

Answer on a scale of 0 to 100.
Let 0 = "Impossible to end in a draw"
50 = "Equally likely to end in a draw or not",
100 = "Definitely going to end in a draw"

You may first write out your thoughts on a scratchpad.
When you feel you understand the game and are ready to respond, provide a single number
between 0 to 100. Write your responses as a number, in the form RESPONSE-Q1 =
<your-numerical-response-to-q1> and RESPONSE-Q2 = <your-numerical-response-to-q2>
```

**Funness evaluation prompt, for an example game**

```
Imagine you are playing the following game:

Board size: 7 x 7
Win conditions: Each player needs 4 pieces in a row to win. The first player can only
win by making a diagonal row, but the second player does not have this restriction.

How fun is this game?

Answer on a scale of 0 to 100.
Let 0 = "The least fun of this class of grid-based game"
50 = "Neutral"
100 = "The most fun of this class of grid-based game"

You may first write out your thoughts on a scratchpad.
When you feel you understand the game and are ready to respond, provide a single number
between 0 to 100. Write your response as a number, in the form RESPONSE =
<your-numerical-response>
```

## A3.4   ALTERNATE MODELS

We also compare to a series of alternate models implemented in  (Collins et al., 2025). We compared against the "Intuitive Gamer," a computational cognitive model which captures how people reason

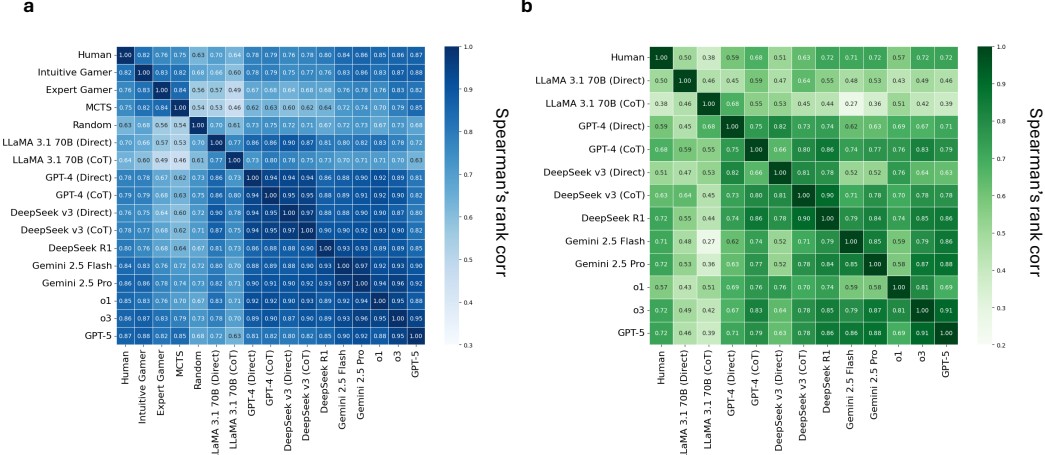

Figure 5: **Spearman's rank correlation over models' and people's predicted payoff and funness judgments.** Rank correlation is computed for **a,** payoff and **b,** funness predictions. Higher means the ranked order of the predicted game evaluation is more similar.

about new games before any experience. The model posits that people engage in fast, flat (depth-limited) goal-directed probabilistic reasoning. The model can be scaled up toward a more sophisticated "Expert Gamer" model which implements deeper tree search inspired by the depth-5 model in van Opheusden et al. (2023). We also compared against Monte Carlo Tree Search (MCTS) (Coulom, 2006; Genesereth and Thielscher, 2014; Silver et al., 2016) and random agents, examples of player agents with greater and lesser sophistication. We only compare against these alternate models for the payoff predictions, as the funness models are regression models fit to a subset of the human data, rendering the comparison less clear. We refer to Collins et al. (2025) for details on all alternate models.

## A4 ADDITIONAL ANALYSIS DETAILS

We include additional details into model evaluations, based on the estimated game-theoretic payoffs and further comparisons to human evaluations of payoff and funness.

In addition to the $R^2$ over payoff and funness evaluations reported in the main text, we compute Spearman's rank correlation over the games evaluations (see Figure 5); the rank correlation is less discriminative across models, however, the general trends across model families persist. We show additional individual game-level predicted payoff and funness evaluations in Figures 6 and 7 respectively, and we move beyond aggregate analyses to compare the distribution of people and model judgments in Figure 8.

### A4.1 GAME-THEORETIC PAYOFF ESTIMATES AND ADDITIONAL ANALYSES

Game-theoretic payoffs were computed following (Collins et al., 2025): that is, we mathematically compute the optimal payoffs where possible, and otherwise use the value on games where MCTS converged to $\{-1, 0, 1\}$. This yields 78 of the 121 games. Specifically, for our MCTS-based estimates of the game outcomes, we have an MCTS agent play each game in our dataset 50 times. We report an instance of a "convergence" if all 50 trials of the MCTS agent led to the same outcome for Player 1. Each MCTS agent was run with 1000 iterations. The average duration of each 50-match trial was approximately 4.2k seconds and utilized over 7 million nodes.

### A4.2 ADDITIONAL COMPARISONS TO HUMAN PAYOFF EVALUATIONS

We additionally computed the absolute distance between the expected payoff under each model and people, broken down the category of game (Figures 9- 11). This granular breakdown reveals that,

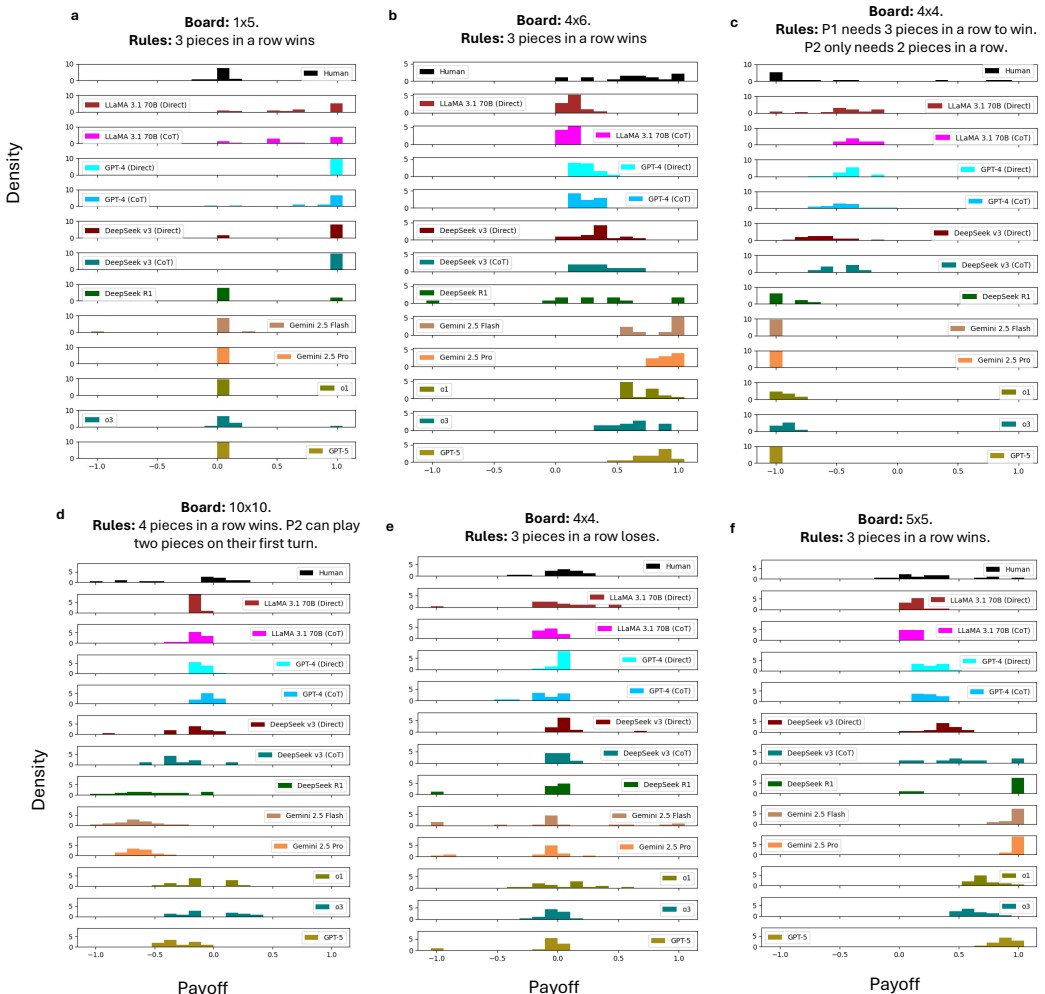

Figure 6: **Distribution over models' and people's predicted payoff judgments for example games.** Example hand-selected representative game evaluations. The distribution over human participants' judgments or each models' 20 rollouts are shown. Panels **a** and **d** show the complete set of models for the examples shown in Figure 2.

even though many reasoning models like OpenAI's o3 better capture human game evaluations in aggregate, there is variability at a per-game level, e.g., for infinite or rectangular boards (Figure 9).

### A4.3 ADDITIONAL COMPARISONS TO HUMAN FUNNESS EVALUATIONS

We compare the absolute deviation in judgments between model-predicted and human-predicted funness at a per-game category level (Figures 12- 14).

.

## A5 ANALYZING REASONING TOKEN USAGE

We include additional analyses into reasoning traces across the reasoning models. Reasoning tokens were extracted from the models' respective APIs, and for DeepSeek-R1, computed using the "DeepSeek-R1-Distill-Llama-70B" tokenizer from the Together AI API for text generated between the "`think`" tokens.

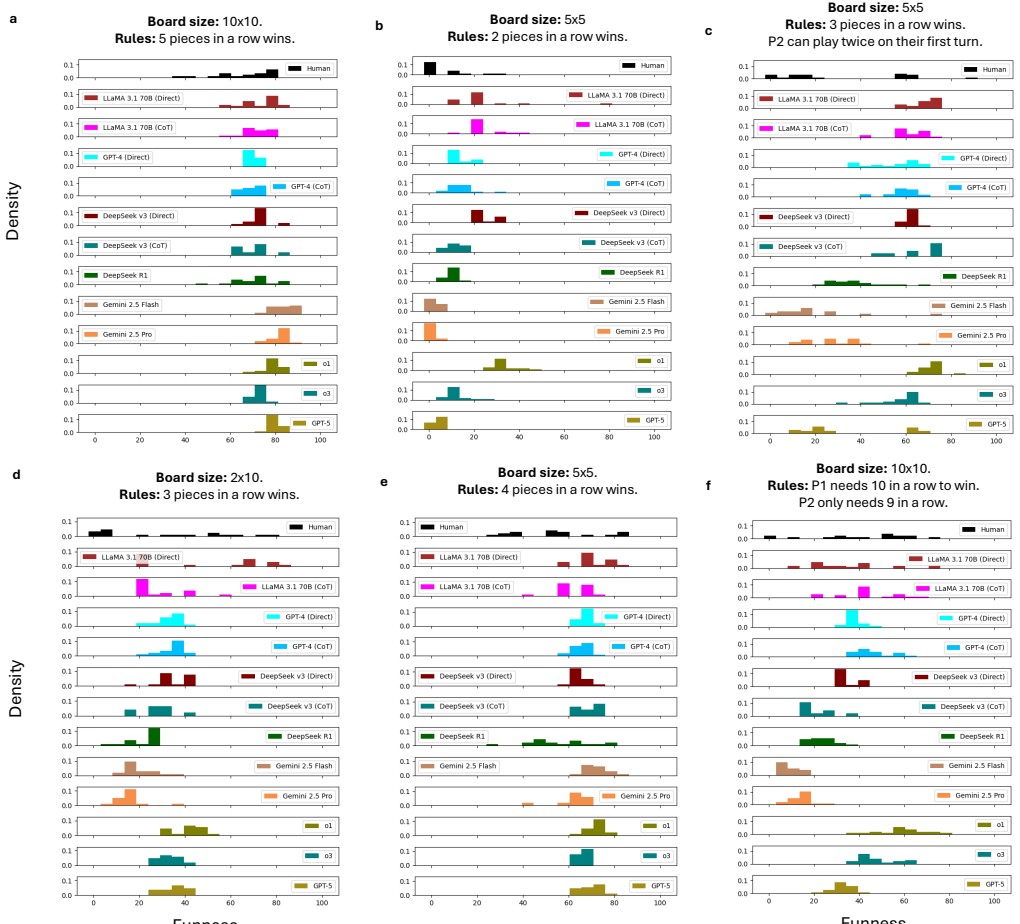

Figure 7: **Distribution over models' and people's predicted funness for example games.** Example hand-selected representative game evaluations. The distribution over human participants' judgments or each models' 20 rollouts are shown. Panels **b** and **c** show the complete set of models for the examples shown in Figure 3.

In exploratory analyses, we find that reasoning token usage is not well-correlated with human judgments, model judgments, or their deviation from the game-theoretic optimal 15. Moreover, preliminary qualitative looks at the content of the reasoning traces (see Section A6.1.1 and A6.3) reveals that while the model can make judgments based on different strategies (e.g., comparing novel games to familiar games such as Connect 4 and proposing features such as first-mover advantage), it still sometimes produces implausible claims or conclusions (e.g., wrongly estimating Player 1 win rate and underestimating the funness of a game).

## A5.1 Varying reasoning amount

Several reasoning models allow users to specify the "amount" of reasoning. In the main text, we reported results using the default ("medium") reasoning threshold. We conducted a preliminary exploration into the impact of varying the reasoning amount specifically for two of the OpenAI family of reasoning models: o3 and GPT-5. There are three options: "low", "medium", and "high". In the main text, we report results using the default ("medium") reasoning threshold. We run a series of exploratory analyses varying the reasoning amount across the "low" and "high" levels. Interestingly, varying the reasoning amount has minimal impact on aggregate fit to human data, but does impact how close to the game-theoretic optimal predictions are (Figure 17). We report games with the highest differences in predicted payoff as a function of reasoning amount in Table 5. The games that

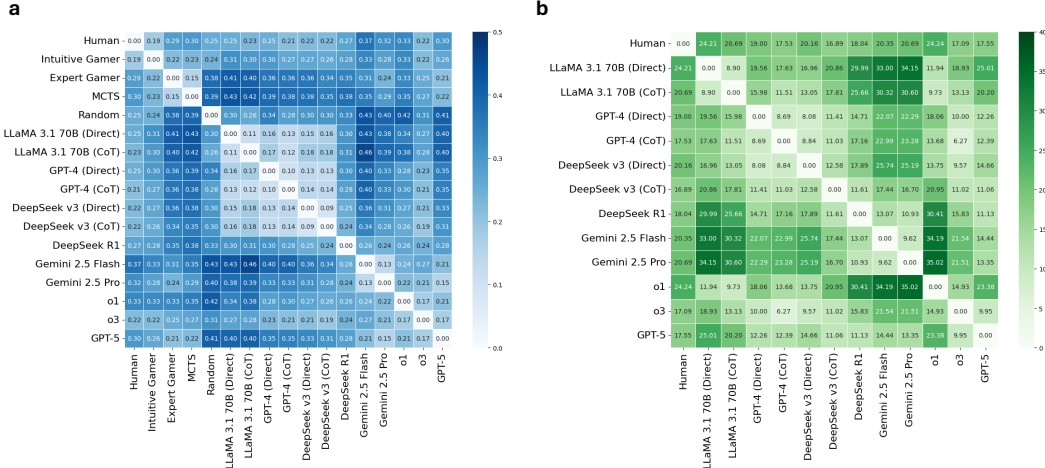

Figure 8: **Comparing distributional alignment of people and models.** Games were judged by approximately 20 people and language and reasoning models were sampled with 20 rollouts per game. The distribution of judgments per game is compared using the Wasserstein Distance (lower means closer) over histograms of judgments per game. Histograms are over the range −1 to 1 for payoff (**a**) and 0 to 100 for funness (**b**).

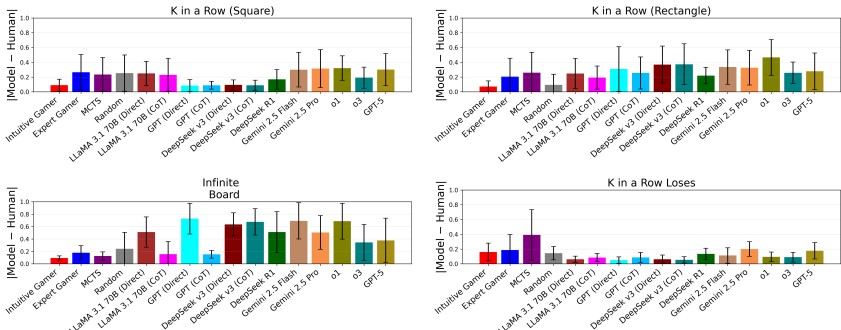

Figure 9: **Distance between model and human payoff predictions, by game category.** Averaged absolute difference between model and human payoff predictions, grouped by game category. Averaged over games within each category. Error bars depict standard deviation over absolute distance between model and human payoff predictions for games within the category. $K$ in a row indicates the number of pieces in a row needed to end the game, where horizontal, vertical, and diagonal all count (as in, e.g., a standard Tic-Tac-Toe game). We separate square and rectangular boards are separated for this setting; other categories mix board shape. Payoff values range from −1.0 to 1.0.

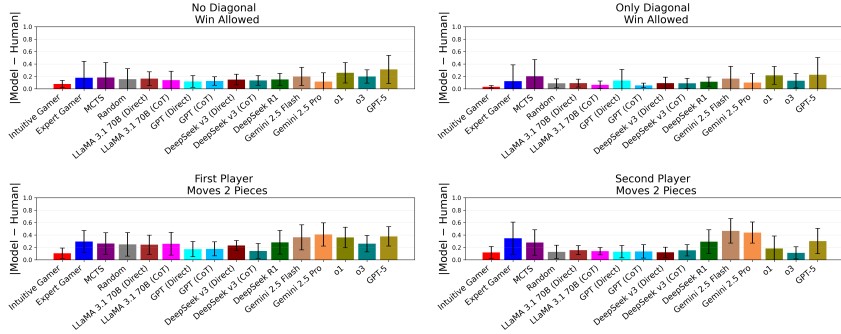

Figure 10: **Distance between model and human payoff predictions, by game category (continued).**

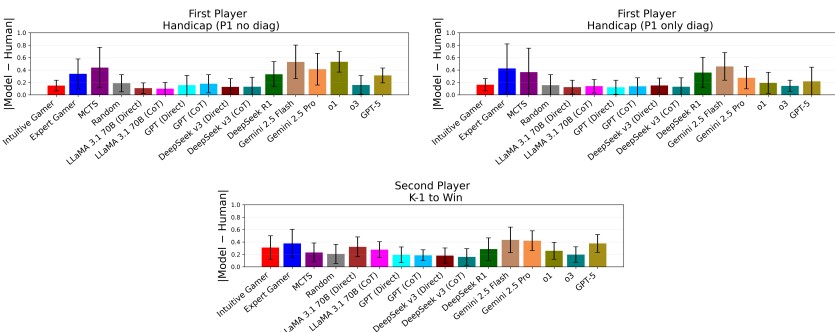

Figure 11: **Distance between model and human payoff predictions, by game category (continued).**

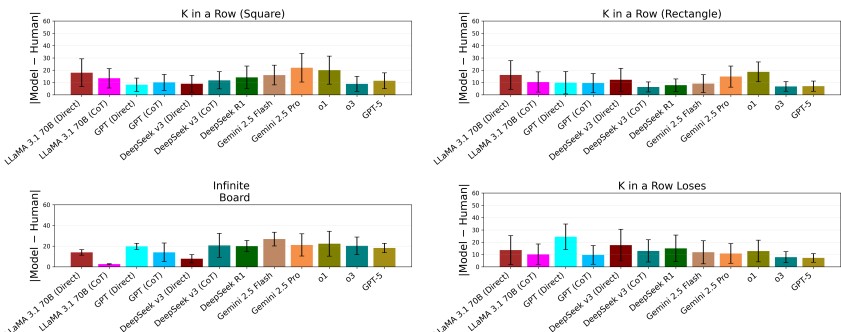

Figure 12: **Distance between model and human funness predictions, by game category.** Averaged absolute difference between model and human funness evaluations, grouped by game category. Averaged over games within each category. Error bars depict standard deviation over absolute distance between model and human funness evaluations for games within the category. Funness values range from 0 to 100.0.

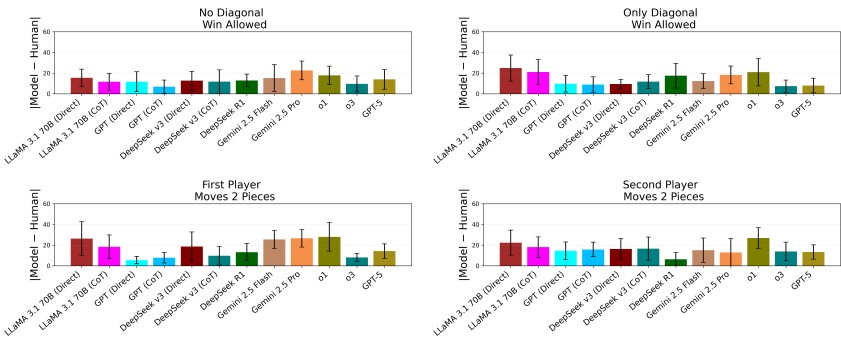

Figure 13: Distance between model and human funness predictions, by game category (continued).

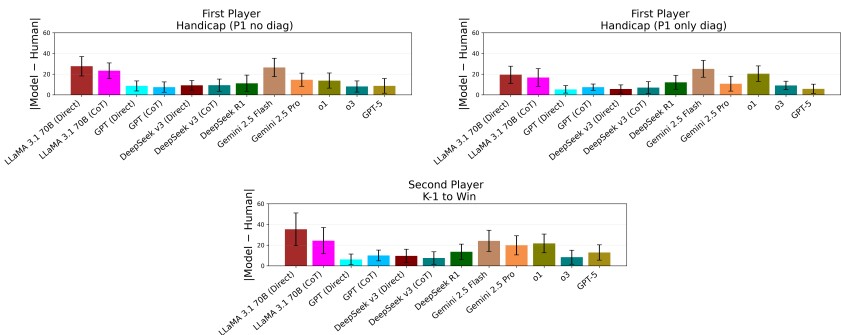

Figure 14: Distance between model and human funness predictions, by game category (continued).

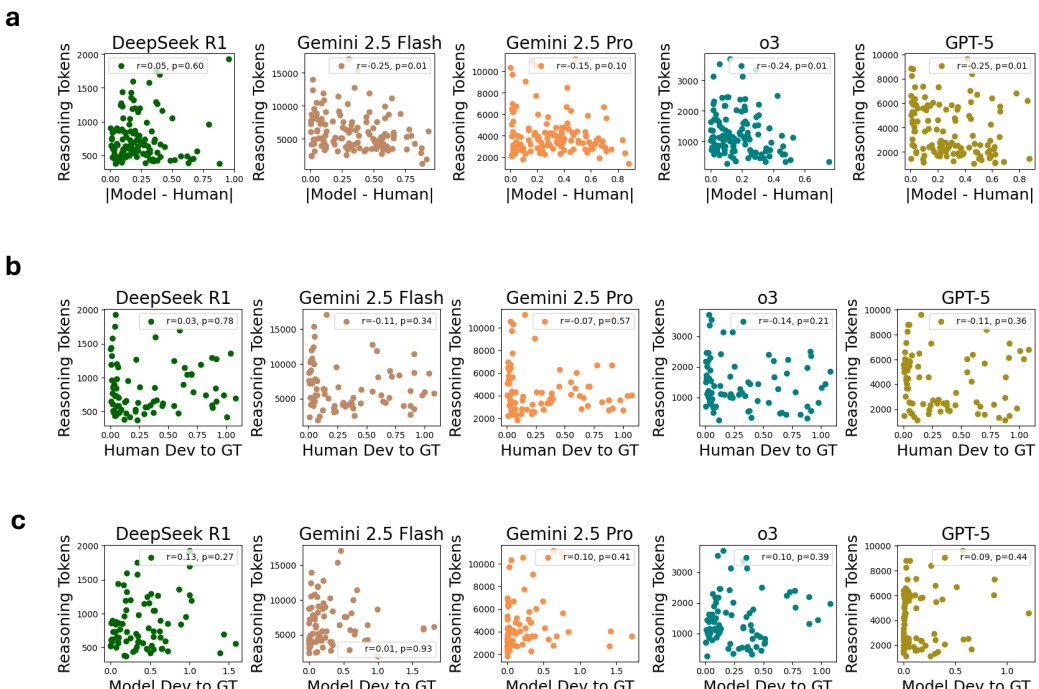

Figure 15: **Reasoning token usage compared to human- and model-estimated payoff. a,** Median usage relative to deviation between model and human; **b,** human and game-theoretic optimal, and **c,** that model and the game-theoretic optimal.

| Reasoning Type | DeepSeek R1 | Gemini 2.5 Flash | Gemini 2.5 Pro |
|---|---|---|---|
| Explicit Simulation | 15.4% / 10.8% | 34.7% / 21.0% | 43.8% / 40.9% |
| Analogical Reasoning | 76.9% / 98.5% | 76.8% / 93.8% | 82.6% / 97.9% |
| Mathematical Computation | 44.8% / 15.0% | 38.1% / 11.6% | 47.0% / 25.6% |

Table 4: **Methods used in reasoning trace for evaluating games.** Reasoning traces are coded based on whether they involve explicit game simulation; analogical reasoning (e.g., comparing the game being evaluated to Tic-Tac-Toe or Gomoku); or mathematical reasoning (e.g., attempting to compute the game-theoretic optimal payoff based on mathematical game properties). Reasoning traces may involve more than one method (or none of the above methods). In each cell (for a model and reasoning method), the first number shows the % for the payoff queries; the second number shows the % for the funness queries.

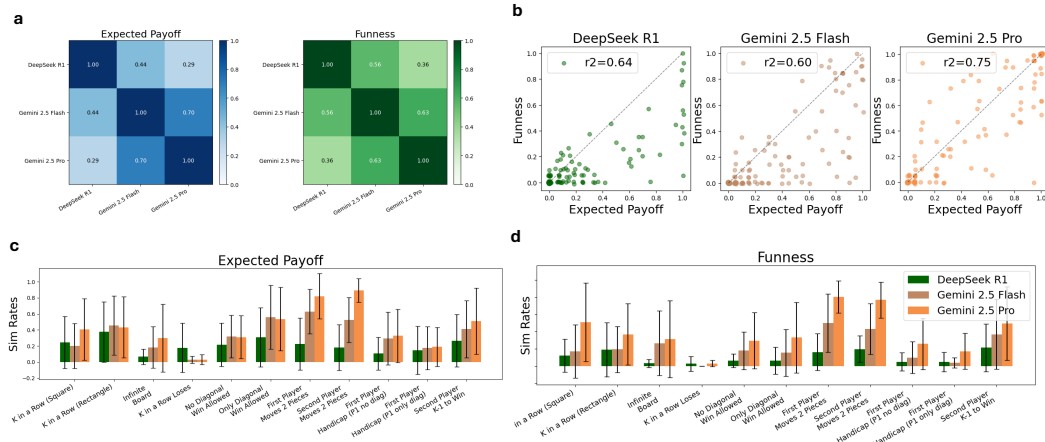

Figure 16: **Explicit game simulation in reasoning models. a,** Correlation ($R^2$) across models in rates of explicit game simulation, across all 121 games; **b,** Simulation rates per game, depending on whether the game was evaluated on fairness (horizontal axis) or funness (vertical axis); **c,** Simulation rates broken down by game category. Error bars depict standard deviation over the simulation rates for the games in those categories.

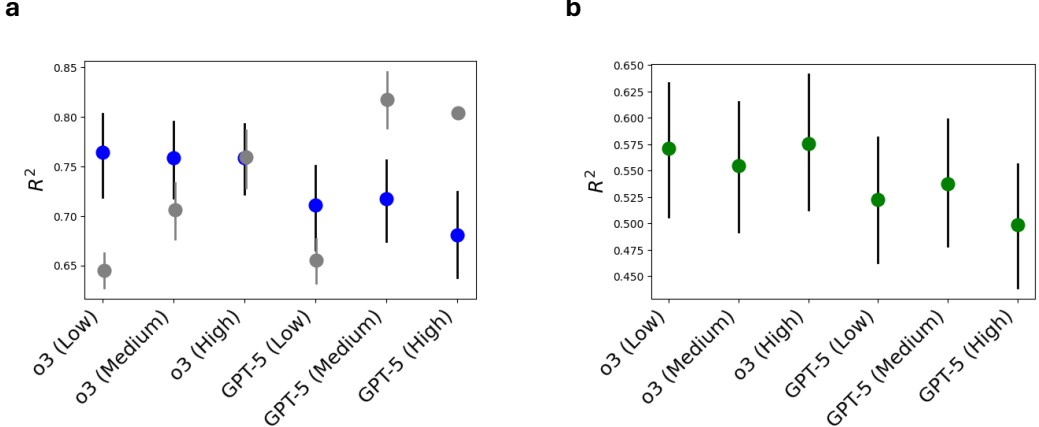

Figure 17: **Assessing evaluations under varied "reasoning amount".** Select reasoning model (o3 and GPT-5) evaluations of games under varied reasoning "amounts". **a,** Bootstrapped $R^2$ relative to people's predicted payoffs (blue) and the estimated game-theoretic optimal (grey). **b,** Bootstrapped $R^2$ relative to people's predicted game funness (green). Error bars depict the bootstrapped 95% CIs over games.

differ most across reasoning amounts generally follow monotonic changes in payoff (to be predicted more or less biased) as a function of the increased reasoning amount. Models are generally fairly consistent across reasoning amounts for some of the simpler games (e.g., where only two pieces in a row are needed to win. But even if all models agree, e.g., $3 \times 3$, 3 pieces in a row wins and the first player gets to go twice on their turn (i.e., "$3 \times 3$ 3 (P1 2p)", the models may not align to peoples' judgements.

## A6 ADDITIONAL DETAILS ON REASONING TRACE CODING

To understand the patterns of reasoning that the reasoning models (for those which we can inspect their produced traces, i.e., DeepSeek-R1 and the Gemini family), we automatically code the content of

| Game | GPT-5 (Low) | GPT-5 (Medium) | GPT-5 (High) | Human |
|---|---|---|---|---|
| 5x5 3 (P1 D) | -0.63 (-0.69, -0.57) | -0.21 (-0.50, 0.11) | 1.00 (1.00, 1.00) | 0.12 (-0.17, 0.41) |
| 10x10 3 (P1 D) | -0.74 (-0.78, -0.70) | 0.47 (0.12, 0.82) | 0.86 (0.86, 0.86) | 0.04 (-0.23, 0.32) |
| 5x5 3 (P1 HV) | -0.29 (-0.40, -0.15) | 0.49 (0.17, 0.75) | 1.00 (1.00, 1.00) | -0.00 (-0.18, 0.20) |
| 4x4 3 (P1 HV) | -0.26 (-0.42, -0.07) | 0.69 (0.44, 0.89) | 1.00 (1.00, 1.00) | -0.08 (-0.30, 0.17) |
| 10x10 3 (P1 HV) | -0.22 (-0.40, 0.01) | 0.50 (0.23, 0.75) | 1.00 (1.00, 1.00) | 0.30 (0.04, 0.57) |
| 10x10 10 P1 / 9 P2 | -0.71 (-0.78, -0.62) | -0.37 (-0.53, -0.22) | -0.08 (-0.08, -0.08) | -0.03 (-0.06, -0.00) |
| 7x7 4 (P2 2p) | -0.34 (-0.39, -0.30) | -0.29 (-0.36, -0.22) | 0.13 (0.13, 0.13) | -0.01 (-0.15, 0.15) |
| 1x10 3 | 0.49 (0.27, 0.71) | 0.02 (0.01, 0.04) | 0.08 (0.08, 0.08) | 0.04 (0.00, 0.10) |
| 3x3 3 L | 0.39 (0.10, 0.65) | 0.30 (-0.00, 0.60) | 0.00 (0.00, 0.00) | -0.03 (-0.09, 0.03) |
| 3x3 3 (P2 2p) | -0.61 (-0.79, -0.42) | -0.58 (-0.76, -0.37) | -1.00 (-1.00, -1.00) | -0.15 (-0.31, 0.01) |
| 10x10 10 D | 0.00 (0.00, 0.00) | 0.00 (0.00, 0.00) | 0.00 (0.00, 0.00) | 0.02 (-0.01, 0.04) |
| 1x5 2 | 1.00 (1.00, 1.00) | 1.00 (1.00, 1.00) | 1.00 (1.00, 1.00) | 0.61 (0.42, 0.80) |
| 1x5 3 | 0.00 (0.00, 0.00) | 0.00 (0.00, 0.00) | 0.00 (0.00, 0.00) | 0.01 (-0.02, 0.04) |
| 10x10 2 | 1.00 (1.00, 1.00) | 1.00 (1.00, 1.00) | 1.00 (1.00, 1.00) | 0.91 (0.79, 0.99) |
| 10x10 3 P1 / 2 P2 | -1.00 (-1.00, -1.00) | -1.00 (-1.00, -1.00) | -1.00 (-1.00, -1.00) | -0.78 (-0.90, -0.65) |
| 3x3 2 | 1.00 (1.00, 1.00) | 1.00 (1.00, 1.00) | 1.00 (1.00, 1.00) | 0.78 (0.64, 0.91) |
| 3x3 3 (P1 2p) | 1.00 (1.00, 1.00) | 1.00 (1.00, 1.00) | 1.00 (1.00, 1.00) | 0.54 (0.39, 0.70) |
| 3x3 3 P1 / 2 P2 | -1.00 (-1.00, -1.00) | -1.00 (-1.00, -1.00) | -1.00 (-1.00, -1.00) | -0.75 (-0.93, -0.54) |
| 4x4 3 P1 / 2 P2 | -1.00 (-1.00, -1.00) | -1.00 (-1.00, -1.00) | -1.00 (-1.00, -1.00) | -0.63 (-0.88, -0.35) |
| 5x5 2 | 1.00 (1.00, 1.00) | 1.00 (1.00, 1.00) | 1.00 (1.00, 1.00) | 0.99 (0.98, 1.00) |

Table 5: **Payoff evaluation differences with varied reasoning amount. Top,** Games with the highest average predicted difference in payoff by GPT-5 under the low versus high reasoning setting. 95% CIs around the mean payoff are reported per model variant. **Bottom,** ten games where GPT-5 variants agreed despite differences in amount of reasoning. For space, abbreviated game names are used, indicating the board size, $K$ in a row to win (e.g., $1 \times 10$ 3 means 3 pieces in a row is needed to win); games where Player 1 requires an extra piece in a row to win (e.g., 3 compared to Player 2 only needing 2) is listed as 3 P1 / 2 P2); games where a player can place twice on their first turn is listed as $2p$; games where a player can only win along a certain direction, like Player 1 only winning diagonally is listed as P1D. And games where the first player to get $K$ in a row loses is denoted with an L.

the reasoning traces, based on the method of computation they involve (explicit simulation; analogical reasoning; mathematical computation) as well as what factors of funness the models discuss as part of their evaluation. We code reasoning traces using o3 (specifically, version o3-2025-04-16, at its default medium reasoning setting and default temperature). Due to the computational cost of coding all 20 sampled traces for all games across many dimensions, we only run o3 once per trace. We therefore caveat that these evaluations are an approximation of the content of the reasoning traces; future work can explore scaling and further verifying reasoning trace analyses.

### A6.1 CODING METHODS EXPLICITLY ENGAGED IN REASONING MODELS' TRACES

Reasoning traces were coded based on the kind of computation they involve. The annotation model (o3) was prompted to respond with a binary YES or NO if the reasoning trace involved either explicit game simulation; analogical reasoning; or mathematical computation (e.g., trying to compute the expected optimal based on features like board size). Each query was asked independently for each rollout and each game. Note, these analyses are over the explicit reasoning traces produced by the models; it is possible they are engaging in other kinds of evaluative methods that is not explicitly written in the reasoning trace, or, that the methods engaged in the reasoning trace are not appropriately accounted for in the final evaluation. Two authors from our team manually inspected several traces for validity; we are actively expanding verification of the coding. Prompts are provided below.

---

**Reasoning trace categorization prompt for assessing analogical reasoning**

```
You are categorizing reasoning traces written by agents reasoning about games.
Your task is to categorize whether the trace involves any explicit analogical reasoning.
That is, the reasoning traces involves a comparison to one or more other games.
Respond with only a single word.
Either:
```

```
YES if it involves analogical reasoning, or
NO if it does not involve any analogical reasoning.
```

---

**Reasoning trace categorization prompt for assessing explicit mathematical calculation**

```
You are categorizing reasoning traces written by agents reasoning about games.
Your task is to categorize whether the trace involves any explicit mathematical
calculations.
That is, particular mathematical operators (+, -, *, /, etc) to assess the question.
The mathematics doesn't need to be correct, it just needs to be explicit calculations.
The mathematical calculations should be precise; just simulating play doesn't count.
Respond with only a single word.
Either:
YES if it involves explicit mathematical calculations, or
NO if it does not involve any explicit mathematical calculations.
```

---

**Reasoning trace categorization prompt for assessing explicit simulation**

```
You are categorizing reasoning traces written by agents reasoning about games.
Your task is to categorize whether the trace involves any explicit game simulation.
That is, that the trace includes explicit playout behavior for any game, clearly
spelling out who moves on which turn.
Categorize whether the following includes explicit playout simulation.
The simulation doesn't need to be correct, nor go to the end of the game. But it needs
to involve turn taking and move selection.
Respond with only a single word.
Either:
YES if it involves explicit playout simulation, or
NO if it does not involve any explicit simulation.
```

### A6.1.1 EXAMPLE REASONING TRACES ACROSS EVALUATIVE METHODS

We include three example traces demonstrating example evaluative methods.

---

**Reasoning trace from DeepSeek-R1 engaging analogical reasoning**

```
-------------------------------------------------
10x10
first player to get 9 in a row wins.
-------------------------------------------------

Alright, let me think through this game scenario. We have a 10x10 board where the first
player to get 9 in a row wins.

First, considering the win condition: 9 in a row is quite long on a 10x10 grid. The
board is large enough that blocking attempts become more complex than in smaller games.
However, achieving 9 uninterrupted pieces seems difficult when the opponent can actively
block.

For Q1 (first player win likelihood if not a draw): Given the board's spaciousness, the
first player might have an initial advantage in claiming space. But the second player
can effectively mirror or counter strategies. I recall that in similar games like Gomoku
(5-in-a-row on 15x15), professional play often leads to draws with perfect play. Here,
9-in-a-row is even harder to achieve, making decisive wins unlikely. I'd estimate the
first player's win chance at just 10\% - only possible if the opponent makes significant
errors.

For Q2 (draw likelihood): The large board and high win threshold make it extremely
difficult for either player to force a win against optimal defense. This resembles
high-level Go or chess endgames where draws dominate. Unless one player blunders, the
game should end in a draw nearly every time. I'd put draw probability at 90\%.
Both estimates lean heavily toward draws due to the balanced nature of the setup and the
difficulty of achieving such a long connection.
```

---

**Reasoning trace from DeepSeek-R1 involving explicit simulated play**

```
-------------------------------------------------
2x10

3 pieces in a row wins
-------------------------------------------------
```

Alright, let me tackle this problem step by step. The game is on a 2x10 grid, and the goal is to get three pieces in a row. Both players play reasonably, and I need to figure out the likelihood of the first player winning (if not a draw) and the chance of a draw.

First, the board is narrow with only two rows. Getting three in a row horizontally would require placing three consecutive pieces in the same row. Vertically, since there are only two rows, getting three in a vertical line is impossible. So winning must be horizontal.

Each player takes turns placing pieces. The board has 20 cells. The first player starts, and they'll aim to block each other. Given the narrow rows, making three in a row might be tricky because opponents can easily block. For example, if the first player places a piece in row 1, column 1, the opponent can place adjacent to block the potential three.

Considering the board length (10 columns), there's room to create multiple threats. But with only two rows, players can cover both rows quickly. Maybe neither player can force a win if both play optimally. Every time someone tries to start a sequence, the other can block. Since the board is long, but narrow, it might lead to a situation where all spots are filled without a three-in-a-row.

Total moves possible are 20. Even if players aim to block, if the game continues to the end without three in a row, it's a draw. So perhaps a draw is very likely. If both players are playing perfectly, they'll block each other's attempts, leading to a filled board with no winner. Therefore, the chance of a draw is high.

For Q2, I'd say around 90-95 because it's almost certain if both play well. For Q1, since the draw is highly likely, the chance of the first player winning if not a draw is low. But if not a draw, say 10\% chance the first player wins? But actually, if there's no draw, it means someone wins. Since both players are symmetric except for going first, maybe the first player has a slight advantage. But in such a tight board, perhaps it's 50-50? But considering that the second player can always block, maybe if a win happens, it's equally likely for either player. So Q1 around 50.

But let me think again. The first player moves first, so they might create the first threat. But the second player can block and counter. Given the board size, it's possible that the first player can set up a fork or something. However, with only two rows, it's hard to create multiple threats simultaneously. Each move might be blocked immediately.

After deeper consideration, it seems almost impossible for either player to force a win without the other making a mistake. Therefore, the game will almost always end in a draw. So Q2 is 100? Or close to 100. Let's say 98. For Q1, since draws are so common, but if there's a win, it's 50-50 who gets it. So Q1 at 50.

But the question is \"if the game does not end in a draw\", so when there's a win, how likely the first player wins. Given symmetry, it should be 50. But perhaps the first move gives an advantage, so maybe 55? However, in this constrained board, the advantage might be minimal.

Looking at known similar games. Tic-tac-toe on 3x3 is a draw with perfect play. Here, the board is 2x10, which is longer but narrower. Getting three in a row horizontally is harder because the opponent can always block the third piece. For example, if Player1 places in (1,1), Player2 in (1,2), then Player1 can't place in (1,3) without Player2 blocking earlier. Actually, Player2 can always block the immediate threats.

Therefore, the game will end in a draw with optimal play. So Q2: 100, but maybe not exactly 100 because if a player makes a mistake, but assuming perfect play, it's always a draw. So Q2 is 100. Then Q1: since no wins occur, the condition \"if not a draw\" is impossible, so it's undefined. But the question requires a number. In such cases, perhaps the likelihood is 50 as a default? Or maybe 0? However, the question says \"if the game does not end in a draw\", which implies that we're conditioning on there being a winner. If both play perfectly, there's never a winner, so that scenario doesn't happen. Hence, the answer for Q1 is irrelevant, but mathematically, it could be 50 as a neutral value. But in reality, with perfect play, it's a draw 100\% of the time. So for Q2, 100. For Q1, since there are no wins, but the question is conditional is on \"if not draw\", which never happens. So perhaps Q1 can be set to 50 arbitrarily.

Alternatively, considering that if a win occurs (which it shouldn't with perfect play), it's equally likely, so 50. So I'll go with Q1: 50, Q2: 100.

But let me double-check. Suppose the first player tries to start at position (1,1). Second player plays at (1,2) to block. Then first player plays at (2,1), second at (2,2). Then first plays at (1,3), second blocks at (1,4), and so on. This continues without a three-in-a-row. The entire board fills without a win. So yes, draw every time.

Therefore, Q2: 100 (definitely draw). Q1: 50 (if not draw, equally split, though it's impossible). So the answers are 50 and 100.

---

**Reasoning trace from DeepSeek-R1 engaging mathematical reasoning**

---

```
-------------------------------------------------
5x10 grid, and the win condition is getting 6 pieces in a row
-------------------------------------------------
```

Alright, let's tackle this problem step by step. The game is on a 5x10 grid, and the win condition is getting 6 pieces in a row. Both players are playing reasonably, and I need to estimate two things: the likelihood of the first player winning if it's not a draw (Q1) and the probability of a draw occurring (Q2).\n\nFirst, considering the board dimensions. A 5x10 grid has 50 cells. The win requires 6 in a row. On a board that's 5 rows high, horizontally, the maximum consecutive spaces in a row would be 10. So there's space for potential winning lines. Vertically, since it's only 5 rows, you can only get 5 in a column, which isn't enough for a win. Therefore, no vertical wins are possible in this setup. So winning lines can only be horizontal or diagonal.\n\nNow, diagonals. For diagonals, the length possible depends on the direction. For diagonals going from top-left to bottom-right, the maximum length in a 5x10 grid would be limited. For example, starting from the top-left, a diagonal could go down 5 rows but only across as many columns as the row allows. Actually, in a 5x10 grid, the longest diagonal would be 5 cells. Wait, 5 rows, so diagonally, the longest possible is 5 cells. For example, from (1,1) to (5,5) is 5 cells. That's not enough for 6 in a row. Similarly, other diagonals would also max out at 5 cells. Therefore, diagonal wins are impossible in this 5x10 grid because the board isn't tall enough to get 6 in a diagonal. So the only possible way to win is horizontally.\n\nSo players can only win by placing 6 consecutive pieces in the same row. Since each row has 10 cells, it's possible to have multiple potential 6-in-a-row lines per row. For example, each row has 5 possible winning lines (positions 1-6, 2-7, 3-8, 4-9, 5-10). With 5 rows, that's 25 possible winning lines.\n\nNow, the game is similar to Connect 4 or Gomoku, but with the constraint that only horizontal wins are possible. This might make it easier to block potential wins. The first player starts by placing a piece, then the second player responds. Since the board is wider (10 columns) but only 5 rows, players will fill up the rows over time.\n\nGiven that both players play reasonably, they will try to create their own winning lines while blocking the opponent. The first player might have a slight advantage by taking the initiative. However, the board being 5 rows high might mean that it's harder for the first player to force a win because the second player has more opportunities to block.\n\nConsidering the total number of moves. The board has 50 cells. Each player places one piece per turn. If all cells are filled without either player getting 6 in a row, the game is a draw. The question is whether a draw is likely.\n\nIn games like Gomoku on a larger board, draws are less common because players can form lines, but here the win condition is 6 in a row on a board that's only 5 high. Since vertical and diagonal wins are impossible, blocking is easier. The second player can always respond to threats by placing a piece to block the first player's potential lines.\n\nMoreover, with 5 rows, once a row has several pieces, players might switch to another row, spreading out the moves. This might lead to the board filling up without any player achieving 6 in a row. Since there are 10 columns per row, it's possible that rows get filled in such a way that no 6 consecutive cells are free for a win. For example, if pieces are placed such that every potential 6-cell line in a row has at least one block from each player, preventing either from getting 6 in a row.\n\nSo, the likelihood of a draw might be higher. Conversely, the first player might still have an advantage by starting in the center of a row, trying to build threats that force the second player to respond. But with the second player able to block multiple rows, the advantage might be minimal.\n\nConsidering known games. Tic-tac-toe on a 3x3 grid always ends in a draw with perfect play. Connect 4 on a 6x7 grid is a first-player win. Gomoku (5 in a row) on a 15x15 board is a first-player win. But here, the board size is smaller in one dimension (only 5 rows), and the win condition is 6 in a row. Since vertical and diagonal wins are impossible, the game is essentially constrained to horizontal play only. This might make it more drawish because players can easily block each other's horizontal lines.

Therefore, the probability of a draw could be high. For Q2, the likelihood of a draw might be around 70-80\%. For Q1, if the game is not a draw, the first player's chance might be slightly above 50\%, but since draws are common, the actual win probability might be low.\n\nBut let's adjust. If both players play perfectly, and the game can be forced to a draw, then Q2 would be high. The second player can always mirror the first player's moves in a different row to prevent any 6 in a row. For example, if the first player places a piece in row 3, column 5, the second player places in row 2, column 5, blocking vertical (though vertical doesn't matter) but also preventing horizontal build-up. Actually, since wins can only be horizontal, players can focus on blocking rows.\n\nEach row independently can be seen as a smaller game. However, players can choose which row to play in. The first player could try to create multiple threats in different rows. But the second player can block those. Since the win requires 6 in a row in one row, and the row has 10 cells, it's possible to create two threats at once. For example, placing a piece that threatens two overlapping 6-in-a-row lines. But with only 5 rows, the second player might be able to block effectively.

Reflecting on total moves. Minimum moves to win: if a player gets 6 in a row in their sixth move. However, the opponent can block. The game might end before all cells are filled if someone gets 6 in a row. But if both players block effectively, the board might fill up without a winner.

Given that, a draw seems plausible. In fact, with perfect play, the second player might force a draw. So for Q1, if not a draw, the first player might still have a slight edge, say 60\%. But overall, draws are likely.

Alternatively, the first player might start in the center of a row. The second player blocks. The first player starts another row. The second player blocks. Eventually, the rows fill, and without vertical or diagonal threats, neither can get 6 in a row, leading to a draw.

So, I'd estimate:

- Q1: If no draw, first player wins about 60\% of the time. But since draws are frequent, their overall win rate is low.
- Q2: Draw probability is high, say 80\%.

But wait, the question is about the likelihood when the game doesn't end in a draw (Q1) and the likelihood of a draw (Q2). So for Q1, the conditional probability: given that there's a winner, how likely is it that the first player wins? That might be closer to 50-60\% because the first player has the initiative. But in scenarios where a win is possible, the first player might seize it slightly more often. However, with the board constraints, it might be balanced.

Alternatively, perhaps the second player can always ensure that if there's a win, it's theirs. But that seems less likely. So I'll go with 60 for Q1 and 80 for Q2. That is, if the game isn't a draw, the

```
first player wins 60\% of the time, but there's an 80\% chance the game ends in a draw. Therefore, the
actual win rate for the first player would be (1-0.8)*0.6 = 0.12, or 12\%, which seems plausible given
the setup.
```

## A6.2   CODING FUNNESS MEASURES

Additionally, traces produced from evaluating the funness of games are coded based on what measures of the funness a model considered as part of its evaluation. We code whether models discussed the game balance (fairness); strategic richness; challengingness; game length; and/or game novelty. Each query was again asked independently for each rollout for each game. In addition to coding the reasoning traces, we also run o3 over the chain-of-thought rationales written from a selection of the non-reasoning language models.

**Reasoning trace categorization prompt for coding whether balance was considered when assessing funness**

```
You are categorizing reasoning traces written by agents reasoning about games.
Your task is to categorize whether the trace considers game balance when assessing
funness.
That is, whether the trace makes any mention of the fairness of the game, whether the
game is lopsided, whether a player has an advantage, etc.
Respond with only a single word.
Either:
YES if it mentions game balance, or
NO if it does not mention game balance
```

**Reasoning trace categorization prompt for coding whether challengingness was considered when assessing funness**

```
You are categorizing reasoning traces written by agents reasoning about games.
Your task is to categorize whether the trace considers game challenge when assessing
funness.
That is, whether the trace makes any mention of the relative challengingness or ease of
the game.
Respond with only a single word.
Either:
YES if it mentions game challengingness, or
NO if it does not mention game challengingness
```

**Reasoning trace categorization prompt for coding whether game length was considered when assessing funness**

```
You are categorizing reasoning traces written by agents reasoning about games.
Your task is to categorize whether the trace considers game length when assessing
funness.
That is, whether the trace makes any mention of how long the game is expected to take.
Respond with only a single word.
Either:
YES if it mentions game length, or
NO if it does not mention game length
```

**Reasoning trace categorization prompt for coding whether strategic richness was considered when assessing funness**

```
You are categorizing reasoning traces written by agents reasoning about games.
Your task is to categorize whether the trace considers whether a game is strategically
rich when assessing funness.
That is, whether the trace makes any mention of how much strategy the game involves
(e.g., how much strategic depth it has).
Respond with only a single word.
Either:
YES if it mentions the strategic richness of a game, or
NO if it does not mention the strategic richness of a game
```

**Reasoning trace categorization prompt for coding whether game novelty was considered when assessing funness**

```
You are categorizing reasoning traces written by agents reasoning about games.
```

```
Your task is to categorize whether the trace considers game novelty when assessing
funness.
That is, whether the trace makes any mention of how novel the game.
Respond with only a single word.
Either:
YES if it mentions game novelty, or
NO if it does not mention game novelty
```

## A6.3   ADDITIONAL EXAMPLE REASONING TRACES

We provide several additional example reasoning traces, which we selected to demonstrate notable reasoning patterns in model evaluations; here, all traces again from DeepSeek-R1.

---

**Ex 1, Unreasonable fairness judgment from DeepSeek-R1**

```
-----------------------------------------------
Board size: 10 x 10
Win conditions: 7 pieces in a row wins.
-----------------------------------------------
[think]
Okay, the user is asking me to evaluate a specific board game scenario. They've
described a 10x10 grid where winning requires getting 7 pieces in a row. My task is to
estimate two probabilities: the first player's win chance (Q1) and the draw probability
(Q2), assuming optimal play.

This is an interesting combinatorial game theory problem. I recall that in larger grid
games with longer win conditions, the first-player advantage diminishes compared to
smaller boards. For a 10x10 board with 7-in-a-row, the game has more space and
flexibility.

Considering Q1: In games like Gomoku (5-in-a-row on 15x15), first player wins are
common, but that's a more constrained setup. Here with only 7 needed on 10x10, the board
is relatively smaller for the win condition. With perfect play, the second player can
likely force a draw by mirroring or blocking effectively. I'd estimate first player win
chance below 10\% - so I'll go with 5.

For Q2: Draws become highly probable in such balanced setups. The board has exactly 100
spaces (even number), and with the win requirement being high relative to board size,
exhaustion of moves without a win seems very possible. I'd put draw probability at
around 95\% - both players can prevent wins indefinitely.

The user seems to be conducting cognitive science research, probably studying human
intuition about game symmetries. They're likely looking for reasoned estimates rather
than precise calculations. My responses should reflect the inherent draw-favoring nature
of this configuration.
[think]
After analyzing the game mechanics on a 10x10 grid with a win condition of 7-in-a-row:

**RESPONSE-Q1 = 5**
*(First player has a very low win chance with optimal play due to the board size and win
condition favoring defensive strategies)*

**RESPONSE-Q2 = 95**
*(Draw is extremely likely as the large board and high win threshold allow both players
to consistently block winning formations)*

### Reasoning:
1. **Win Condition vs. Board Size**:
   - 7-in-a-row on a 10x10 grid requires significant space, making early wins unlikely.
   - Second player can mirror moves or block threats effectively (similar to *Gomoku*
theory on larger boards).

2. **Draw Dominance**:
   - Total positions (100) vastly exceed minimum moves to win (13-14 per player).
   - With optimal blocking, neither player can force a win before the board fills.
   - Proven draw strategies exist in comparable games (e.g., 15x15 Gomoku with 5-in-a-row
is a draw with perfect play).

3. **First Player Disadvantage**:
   - No known strong opening for 7-in-a-row on this size.
   - Second player equalizes quickly, reducing P1's theoretical edge to near zero.

*Note: Computational game theory confirms draw-heavy outcomes for similar
configurations.*
```

**Ex 2, Underestimate fun from DeepSeek-R1**

```
------------------------------------------------
Board size: 4 x 9
Win conditions: 4 pieces in a row wins.
------------------------------------------------
```

[think]
Alright, I need to evaluate how fun this 4x9 grid game is where you win by getting 4 in a row. First, I should think about the board dimensions. A 4x9 grid is wider than it is tall, which might affect strategy. Since it's 9 columns wide and only 4 rows high, placing pieces strategically to block opponents while aiming for a row of four becomes crucial.

Hmm, the win condition is 4 in a row. That's similar to Connect 4, but Connect 4 usually has 6 rows and 7 columns. Here, with only 4 rows, vertical stacks might be limited. Players can only stack up to 4 pieces in a column, which is exactly the win condition. So, if someone fills a column, they automatically get a vertical win. But since each column can only hold 4 pieces, and the board is 4 high, that means each column can be completely filled. However, once a column is full, no more pieces can be placed there.

Wait, does that mean every column could potentially be a win for the player who places the fourth piece in that column? But that would make the game very short and possibly deterministic. Let me think. If a player plays in a column, the opponent might block by playing in the same column, but if they do, the first player can just play elsewhere. Since there are 9 columns, the first player could start in column 1, and if the second player blocks in column 1, the first player moves to column 2, and so on. Eventually, the first player could force a win on the fifth move if they keep moving to new columns, but the second player can also start their own columns.

Actually, both players could be building their own columns. Each player can claim columns by placing their pieces. Since you only need four in a row, and vertical is one way, horizontal and diagonals are also possible. The height of the board is 4, so a horizontal win requires four adjacent columns in the same row. With 9 columns, there's plenty of space for horizontal connections. Diagonals might be trickier but still possible.

I recall that in a similar game like Connect 4, the player who goes first has an advantage, especially on smaller boards. But here, the board is asymmetric (4x9). The narrow height might lead to quicker games because vertical wins are easier to achieve. Each time a player places a piece in a new column, they're one step closer to winning that column. Since the columns are independent in terms of vertical wins, the game might be dominated by who controls more columns.

However, horizontal wins require controlling adjacent columns in the same row. This adds a layer of strategy because players need to balance between building their own columns and blocking the opponent's horizontal connections. For example, if Player 1 starts in the center, Player 2 might need to block horizontally to prevent an easy win. But with 9 columns, there's a lot of room, so it might be harder to block all possible horizontal threats.

The large number of columns (9) compared to rows (4) means there's more space horizontally, which could lead to more complex strategies. Players can create multiple threats across different rows and columns. However, the limited vertical space might make it easier to achieve vertical wins quickly. If a player focuses on a few columns, they could secure a vertical win before the opponent can block.

But blocking vertical wins is straightforward—just place a piece in the same column when the opponent has three. The challenge comes when the opponent has multiple threats. With 9 columns, the board might feel a bit spread out. The game's pace could be fast, with players making quick moves. However, the wider board might also lead to longer games if both players are effectively blocking each other's four-in-a-row attempts.

Another aspect is the possibility of draws. Total cells are 4*9=36. Since two players take turns, if no one wins, the game could end in a draw. However, achieving a 4-in-a-row isn't too hard on this board, so draws might be rare. But if both players are vigilant in blocking, the game might drag on.

Comparing this to the classic Connect 4, which is 6x7, this game has a shorter vertical dimension but wider horizontally. Connect 4 is known for being solvable, where the first player can force a win with perfect play. On a 4x9 grid, the first player might have an advantage due to the odd number of columns and the ability to control the center.

The win condition being only four in a row on a 4x9 grid might lead to less strategic depth. Vertical wins are quick, and horizontal wins require fewer connections than in larger grids. The larger board size horizontally (9 columns) could allow for more complex strategies, but the low height (4) might reduce the complexity since players can't build tall stacks.

Additionally, diagonal wins might be harder to achieve because the board isn't square. Diagonals that span four columns would require four consecutive rows, but since the

```
height is only 4, diagonals are possible both upwards and downwards. For example, a
diagonal starting at row 1, column 1 going to row 4, column 4 is possible. Similarly,
diagonals from row 4 to row 1.

Considering fun, if the game is too simple and the first player can force a win, that
reduces replayability and fun. On the other hand, if there's a good balance between
offense and defense, it could be engaging. The wider board might allow for creative
plays, but the low height might make it feel constrained.

I remember that in games like Gomoku, a larger board allows for deeper strategy, but
here, the 4x9 is not standard. Perhaps the asymmetry adds uniqueness, but it might
unbalance the game. Also, with only four rows, the board fills up faster, leading to
shorter games. Shorter games can be fun for quick sessions but lack depth for prolonged
engagement.

On the fun scale, I'd rate this around 45. It's below neutral because the low height
might make vertical wins too easy, potentially favoring the first player. The wide board
offers some strategic depth, but the limitations on vertical space might hinder complex
strategies. Compared to classic Connect 4, which is more balanced, this variation feels
less fun. So, RESPONSE = 45
[think]

After analyzing the 4x9 grid game with a win condition of 4-in-a-row:

- **Board asymmetry**: The width (9 columns) greatly exceeds the height (4 rows),
skewing strategy toward horizontal/diagonal play. Vertical wins are easily blocked due
to minimal stacking depth.
- **First-mover advantage**: With 9 columns, Player 1 can create multiple unblockable
threats. Perfect play often leads to forced wins, reducing tension. Simulations suggest
~75% win rate for Player 1.
- **Limited depth**: Only 4 rows prevent complex board states. Games resolve quickly
(avg. 8-12 moves), favoring speed over strategy.
- **Blocking efficiency**: Defenders can neutralize threats too easily by "capping"
columns or splitting horizontal lines, causing repetitive stalemates in mid-game.
- **Diagonal constraints**: 4-row height restricts diagonal opportunities (e.g., only 5
valid NE-SW diagonals), simplifying path planning.

**Fun assessment**:
- **Strengths**: Fast-paced, accessible for beginners.
- **Weaknesses**: Predictable outcomes, low strategic replayability, imbalance favoring
Player 1.
Compared to classics like Connect 4 (balanced) or Gomoku (deep strategy), this variant
lacks depth and fairness.

**Rating**: 38/100
*(0 = least fun, 50 = neutral, 100 = most fun)*

RESPONSE = 38
```

**Ex 3, Careful deliberation with DeepSeek-R1 assessing different funness features**

```
-------------------------------------------------
Board size: 10 x 10
Win conditions: Each player needs 5 pieces in a row to win. The first player cannot win
by making a diagonal row (only horizontal and vertical rows count), but the second
player does not have this restriction.
-------------------------------------------------
[think]
We are evaluating a game with a 10x10 board.
 Win conditions:
   - First player needs 5 in a row horizontally or vertically (diagonal doesn't count for
the first player).
   - Second player needs 5 in a row in any direction (horizontal, vertical, or diagonal).

 This asymmetry in rules may lead to an imbalance. The first player is restricted, while
the second player has more ways to win.

 Considerations for fun:
   1. Balance: A fun game should be fair and balanced. If one player has a significant
advantage, it might not be fun.
   2. Challenge: Having restrictions can add challenge, but if too restrictive, it might
be frustrating.
   3. Variety: The second player having more options might make the game more dynamic for
one side.

 Potential issues:
   - The first player is handicapped by not being allowed to win diagonally. This might
make it harder for the first player to win, especially since the second player can block
horizontally and vertically and also build diagonals.
```

– On a 10x10 board, 5 in a row is achievable, but the restriction for the first player
might lead to a situation where the second player has an advantage.

How might this affect fun?
– If the game is unbalanced, it might not be fun because one player might win too
often or the other feels unfairly restricted.
– However, if the handicap is intended to balance an inherent first-player advantage
(like in standard 5-in-a-row), then it might be an attempt to balance, but the rules are
different for the two players.

In standard 5-in-a-row (Gomoku) on a large board, the first player often has an
advantage. This rule tries to reduce that by removing a winning condition for the first
player, but then gives the second player an extra condition (diagonals) without
restriction. This might overcompensate and give the second player an edge.

Fun factor:
– A game that is too predictable (because of imbalance) is less fun.
– Players might feel the rules are arbitrary and unfair, which can reduce enjoyment.

However, note that the game might still be fun if the asymmetry creates an interesting
dynamic. But I suspect that the imbalance might be too great.

Let's compare to a symmetric game (where both players have the same rules) and assign a
fun score relative to that.

Given the asymmetry, I would rate this game as less fun than a symmetric version. But
let's assign a number.

Considering the scale:
0 = least fun of this class
50 = neutral
100 = most fun

I would place this game below neutral because the asymmetry is likely to cause
frustration and imbalance. But it's still a playable game, so not the absolute worst.

Let me think of a symmetric game as a baseline: a standard 5-in-a-row on a 10x10 board
would be around 60-70 (it's a classic, but might be a bit large and slow for some).

This game has a rule that might make it less fun than the symmetric version. So I would
rate it lower.

However, note that the second player's ability to win diagonally might add an element of
surprise and complexity. But the first player might feel unfairly treated.

I think a reasonable score would be around 30-40. Let's say 35.

But let me consider: if the handicap is well-calibrated to balance the game, it might be
more fun. However, without extensive testing, it's hard to say. The rule seems arbitrary
and might be confusing. In the absence of evidence that it balances the game, I'll
assume it causes imbalance.

Therefore, I'll go with 35.

However, note that the problem says: "You can define fun however you wish." So I am
focusing on balance and fairness as important for fun.

Response: 35

But note the instruction: write the response as "RESPONSE = 35"

Let me double-check the rules:
– First player: only horizontal and vertical 5-in-a-row count (no diagonal).
– Second player: horizontal, vertical, and diagonal all count.

This is a significant difference. The first player must only build in two directions and
must watch for the second player building in three. The second player has an extra
winning condition and can also block the first player in all directions.

This imbalance is likely to make the game less fun for the first player and possibly too
easy for the second player? But note that the first player moves first, which is an
advantage. The rule might be intended to counter that. However, it might be too much.

Without knowing the exact balance, we go by the face value of the rule. I think it's a
bit too asymmetric.

Final score: 40 (to account for the possibility that it might be interesting for some
players).

But I think 35 is more accurate.

```
 Alternatively, I can think of it as:

  Symmetric game (both have the same rules) is more fun. So this asymmetric version is
less fun.

  However, if the asymmetry exactly balances the first-move advantage, then it could be
fun. But the problem is that the rules are different, which might be confusing and feel
unfair.

  Therefore, I stick to 35.

 Let me adjust: I think 30 is a bit too low, and 40 is a bit too high. So 35 is in the
middle.

 So RESPONSE = 35
[think]
After analyzing the game:

- **Board size (10x10)** allows strategic depth but may feel large for a 5-in-a-row game.
- **Asymmetric win conditions** create imbalance:
  - **First player** is restricted to horizontal/vertical wins only (no diagonals),
limiting options.
  - **Second player** has unrestricted wins (horizontal/vertical/diagonal), granting a
significant advantage.
- **Balance issues**: The first player's handicap combined with the second player's
flexibility likely leads to frustration, predictable outcomes (second player favored),
and reduced fairness.
- **Fun impact**: Asymmetry feels arbitrary and punitive rather than strategic. While
novelty might intrigue initially, long-term engagement would suffer.

Compared to symmetric grid games (e.g., Gomoku, Tic-Tac-Toe), this design sacrifices
balance and fairness-key to fun. It ranks low but avoids being "least fun" due to its
playable structure.

**RESPONSE = 30**
*(Below neutral: Imbalance overshadows potential depth)*
```

## A7   PROMPT SENSITIVITY ANALYSES

We next conduct an initial analysis of prompt sensitivity, focusing on the payoff predictions. We first assess the impact of changing the "kind" of play the model should simulate (reasonable versus optimal) and then explore the impact of flipping the question order.

### A7.1   ASSUMPTION OF "REASONABLENESS"

Participants, as well as models, were instructed to estimate the expected outcomes assuming both players played "reasonably." But, defining and being able to simulate what counts as "reasonable" play is itself somewhat hard to quantify. We conduct an initial analysis of a subset of the GPT family of models' predictions of the expected outcomes of games when the assumption of reasonable play is replaced with "optimal" play. That is, models are asked " assume both players play optimally" rather than "assuming both players play reasonably".

When models are prompted to instead assume players play optimally, the fit of all models tested relative to people drops (see Figure 18a). However, on the games where one can estimate a game-theoretic optimal value, neither o1 nor GPT-5 substantially improves their closeness to the optimal value. In contrast, o3's predictions become comparably close to the game-theoretic optimal as GPT-5 (see Figure 18b-c). This suggests that o3 is more adept at tailoring its response to the questions based on a more human-like assumption of what "reasonable" may mean and that the model is capable of approaching a calibrated estimate of the game-theoretic optimal value of games. In contrast, o1 struggles under the regimes tested here to estimate the game-theoretic optimal value; and conversely, GPT-5 struggles to estimate human-like responses that are sub-rational relative to the game-theoretic optimal. We see an investigation of the controllability of the precision and character of such evaluations as ripe grounds for future work (e.g., one may imagine advantages to being able to simulate how a less rational agent may engage in a new job or on a new math problem). Our findings corroborate other work into the challenges of sophisticated models' assumptions of human rationality (Liu et al., 2025a).

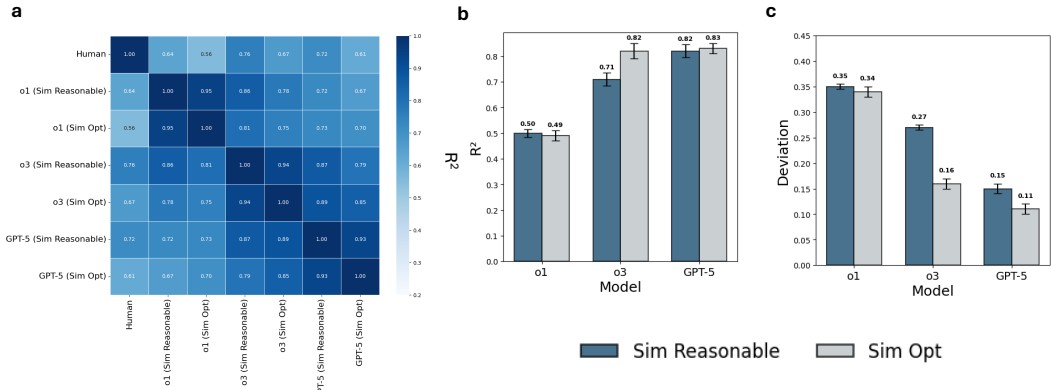

Figure 18: **Impact of simulating "reasonable" vs. "optimal" play in predictions. a,** Model-model and human-model correlation ($R^2$) in predicted payoff over the 121 games, depending on whether models were prompted to assume players player reasonably or play optimally. **b,** $R^2$ relative to the game-theoretic optimal predicted payoff depending on whether models were prompted to estimate payoff based on reasonable versus optimal play. **c,** Absolute difference in predicted payoff relative to the game-theoretic optimal payoff. Error bars for **b** and **c** depict bootstrapped confidence around the mean.

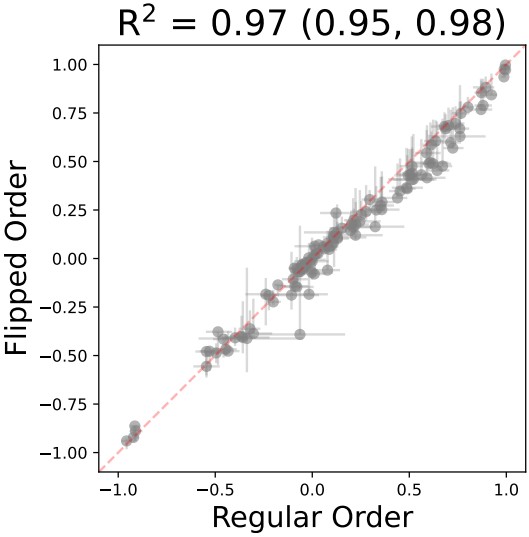

Figure 19: **Impact of question order on payoff judgments.** Average payoff predictions for the 121 games for o3 (with medium reasoning amount) for the original question order (horizontal axis; predicting the first player's likelihood of winning if no draw, then draw likelihood) and a flipped order (vertical axis; predicting draw likelihood before likelihood of player one winning if no draw). Error bars depict 95% CI bootstrapped mean predictions.

## A7.2 QUESTION ORDER

For expected outcome questions (from which payoff was computed), participants as well as models were all instructed to (1) assess the probability that the first player wins given the game does not end in a draw and (2) estimate the probability a match would end in a draw. The order of the questions was fixed (Question 1 was always presented before Question 2). As o3 was the most human-aligned model, we conduct an initial sensitivity analysis into the impact of flipping the order of the questions (presenting the draw probability estimation before estimating the probability the first player wins if no draw). There is little difference in the resulting payoff predictions for o3 (see Figure 19). While

this is not a guarantee that people will not be sensitive to the order, nor other language models, it lends some credence to the potential robustness of our results to question order.

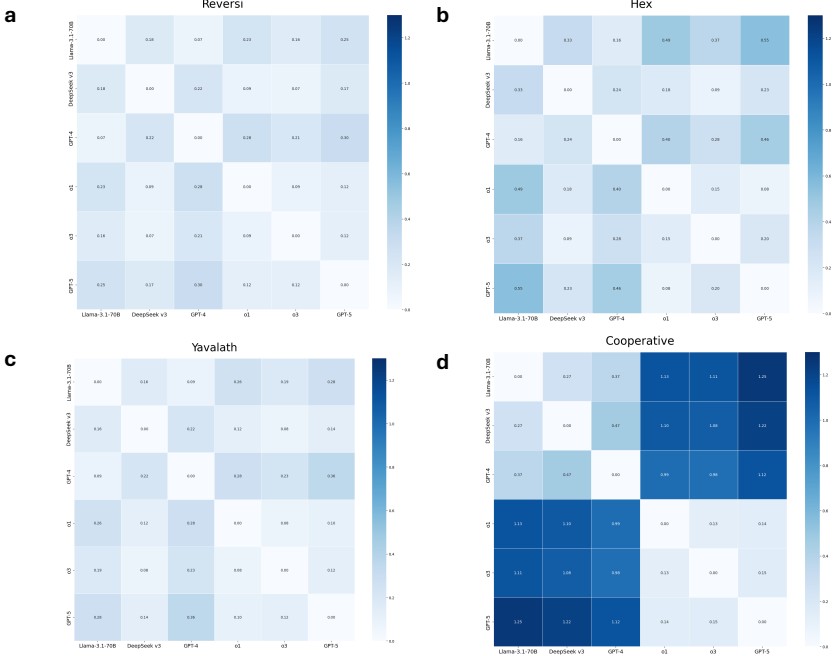

Figure 20: **New game variants.** Averaged mean absolute error in payoff predictions for new game variants. Errors are averaged over all game variants for a game category, over 20 rollouts per LM. Darker blue means more different (higher absolute error). Lighter means more similar.

## A8    PRELIMINARY EXPLORATIONS INTO AN EXPANDED GAME SET

While the space of 121 games is highly varied, we conducted an initial exploration into three other competitive game categories (Reversi, Hex, Yavalath) and one cooperative game category ("cooperative tic-tac-toe" wherein players either both win if they make their pattern or both lose). We developed 15 variants off a base game for each (totaling 16 variants for each of the four game categories). For the competitive variants, we varied board size and turn dynamics (e.g., wherein Player 2 could play twice on their first turn). For the cooperative variant, we varied board size and the patterns (e.g., $4 \times 5$. P1 needs 4 in a row vertically and P2 needs 4 in a row horizontally for both to win).

We ran a subset of non-reasoning language models (LLaMA 3.1 70B; DeepSeek v3; GPT-4—all with CoT) and reasoning-based language models (o1; o3; GPT-5). Models were tasked, as before, with predicting the expected payoff of the game (here, under assumed reasonable play). We computed the mean absolute deviation in the payoff predictions, where higher means the payoff predictions are more different.

For the cooperative variants, there are stark differences in the predicted payoff for the reasoning-versus non-reasoning LMs (Figure 20d). The differences are more variable across the other competitive variants. While we generally observe similar payoff predictions amongst the reasoning models for the competitive variants, there are more deviations between the reasoning and non-reasoning models there (and the absolute differences are less stark than the cooperative variants (Figure 20a-c). However, we caution these results are preliminary.

We are actively exploring other non-LM based game reasoners (e.g., MCTS) on these game variants. Future work can also explore the collection and comparison of human data on these novel game variants. Our results could be used to guide which games are most interesting to gather human data on, e.g., prioritizing games with the biggest differences between models.

