# OpenReview forum: "Evaluating Language Models' Evaluations of Games"
_ICLR.cc/2026/Conference — ICLR 2026 Poster_

### Official Review · Reviewer_Ra3f · 2025-10-27

**Soundness:** 2
**Presentation:** 2
**Contribution:** 2
**Rating:** 4
**Confidence:** 5

**Summary:**

This paper proposes a novel AI evaluation paradigm that shifts from assessing problem-solving to evaluating problem evaluation capabilities. The study uses 121 novel board games (Tic-Tac-Toe variants) to assess language and reasoning models across two dimensions: expected payoff (fairness) and funness. The authors compare multiple models (GPT-4, DeepSeek-v3/R1, Gemini 2.5, o1, o3, GPT-5) against baselines including human judgments and game-theoretic optimal solutions. Key findings include: (1) reasoning models align better with human judgments than non-reasoning models; (2) a non-monotonic relationship exists between game-theoretic optimality and human alignment; (3) funness evaluations show higher cross-model variability; (4) reasoning token usage is highly unpredictable and weakly correlated with task complexity.

**Strengths:**

1. The paper clearly distinguishes between problem-solving and problem evaluation capabilities, a meta-cognitive perspective with theoretical and practical value. The proposed two-dimensional framework of "difficulty to compute" and "difficulty to quantify" provides a generalizable structure for future research.
2. The experiments compare 7+ models against multiple meaningful baselines: human judgments, game-theoretic optimal solutions, and computational cognitive models (Intuitive Gamer, Expert Gamer, MCTS). The comparison of reasoning versus non-reasoning models enables clear assessment of reasoning's value.

**Weaknesses:**

1. All 121 games are Tic-Tac-Toe variants differing only in board size and rules, representing an extremely narrow subset of the game space. The exclusive focus on two-player, zero-sum, perfect-information competitive board games undermines claims about general "problem evaluation" capabilities.
2. The paper treats funness as a "hard to quantify" query but provides no clear evaluation framework, instead instructing participants "you can define fun however you wish." This lack of standardization means observed human-model misalignment may reflect disagreement about what funness means rather than evaluation capability differences.
3. Game-theoretic optimal values are only computable for 64% of games (78/121), with the remainder relying on MCTS convergence without reported convergence criteria, runtime, or accuracy validation. The key finding of "non-monotonic relationship between optimality and human alignment" depends critically on the correctness of these values.
4. Results are based on a single prompt design with heterogeneous temperature settings (1.0 vs 0.7) and only 20 rollouts per model, despite the authors acknowledging that "model performance is sensitive to factors like exact prompt." Appendix Figure 18 shows reasoning amount significantly impacts results, yet this is not thoroughly analyzed.

**Questions:**

NA

---

> ### Author Response · Authors · 2025-11-20
> **Thank you for your review!**
>
> We thank the Reviewer for recognizing that our paper “provides a generalizable structure for future research” with empirical contributions (e.g., “The comparison of reasoning versus non-reasoning models enables clear assessment of reasoning's value.”). We appreciate your questions, which motivated new experiments that we believe have strengthened our work. We have updated our paper directly (see the latest PDF revision) with new figures and text marked in red. We respond to each of your points in turn.
>
> “All 121 games are Tic-Tac-Toe variants differing only in board size and rules, representing an extremely narrow subset of the game space.”
> - This is indeed a limitation, which we acknowledge in Section 6. We hope more researchers expand the classes of games, and kinds of game queries, when evaluating not just play but the evaluations models make of games before any play!
> However, even though the games can be cast as variants of tic-tac-toe, they do span quite a wide range of mechanics that lead to rich strategic behavior (see Table A2). Some games are so complex that MCTS does not converge, nor is there a known game-theoretic optimal value! Moreover, even though we focus on a single class of two-player competitive strategy games here, we do consider many games within that class – in contrast to many works in AI and psychology which focus on just one game (e.g., Go, Chess, Diplomacy, etc). We see value in having a somewhat constrained yet still rich testbed – where already there are substantial insights to be gleaned into the ways models (and people) evaluate these games before any play! And despite being variants of tic-tac-toe, the games capture many hallmarks of real-world decision making and planning problems.
> - We also offer an analogy to justify the kind and amount of games we study in this work: game evaluation is a kind of task like mathematical reasoning, and one would not expect a good math benchmark (e.g., GSM8K) to cover all kinds of problems at all levels. This may be taken into consideration especially because we are taking one of the first steps towards the game evaluation direction (and to study such work from the perspective of the costs/resources needed in such evaluations).
>
> “The paper treats funness as a "hard to quantify" query but provides no clear evaluation framework, instead instructing participants "you can define fun however you wish."”
> - We are glad you pointed this out! Indeed, the prompt that Zhang et al 2024 and Collins et al 2025 had given to participants did instruct participants to define fun however they wished. And we applied the same instruction to models. This is partly by design. Funness is somewhat ambiguous in how one may define it; as such, when we compare models’ and peoples’ evaluations of funness, it both incorporates some quantification of various questions one may ask when deciding if a game is fun (e.g., is the game suspenseful? Is it challenging? How long is it likely to last?), as well as what those questions/factors are in the first place. As we pointed out to Reviewer cvbH, who raised a similar question, we cannot know exactly what factors people considered from the human data we have access to, but we can take initial steps to try to understand the factors models consider (see Table 2 and Appendix A6). As you note, this is part of the conceptual structure we lay out in our two dimensions.
>
> “Game-theoretic optimal values are only computable for 64% of games (78/121), with the remainder relying on MCTS convergence without reported convergence criteria, runtime, or accuracy validation. The key finding of "non-monotonic relationship between optimality and human alignment" depends critically on the correctness of these values.”
> - Yes, only a portion of the games have estimable game-theoretic values. We follow Collins et al, 2025 in how these were computed. First, one can mathematically compute the game-theoretic optimal value for many of the standard M-N-K variants (i.e., games played on an M by N board where the win condition is whomever gets K in a row first). Then, a second set of games are included if MCTS converges to {-1, 0, 1} after 50 simulations of 1000 tree expansions each. This produces the set of 78 of the 121. We do not have confidence in the MCTS estimate for the remaining games and therefore exclude them from the game-theoretic analysis. We plan to include additional statistics into these runs in the Appendix and will update our paper accordingly; but wanted to share the bulk of our response to you now.

---

> > ### Author Response · Authors · 2025-11-20
> > **Response part 2**
> >
> > (separated due to character limit)
> >
> > “Results are based on a single prompt design with heterogeneous temperature settings”
> > - As described in our response to Reviewers Tg2R and AE1w, we have added a new prompt sensitivity analysis to Appendix A7. And on heterogeneous temperature, all models where we could set the temperature were set to the default (0.7); some models (e.g., some of OpenAI’s reasoning models) did not allow changes to the temperature. However, indeed, the bulk of our study focuses on running models with a single temperature for each model partly for scalability – we opted for running a wider range of models (over all games, and multiple question types). However, we do conduct an initial study into the impact of another hyperparameter (reasoning amount) in Appendix A5 (to which we include new additional analyses, see below).
> >
> > “Appendix Figure 18 shows reasoning amount significantly impacts results, yet this is not thoroughly analyzed.”
> > - Thanks for calling attention to this section! We also find the difference in choice of reasoning amount on model performance (and also, where it does not impact model performance) as interesting. We have added additional analysis in Appendix A5.1 and Table 5 (specifically, surfacing games whose rating changed – and did not change – with varied reasoning amounts).
> >
> > Thanks for your further consideration. We are happy to provide additional details if needed!

---

> > > ### Comment · Reviewer_Ra3f · 2025-11-25
> > >
> > > Thanks for your response. My concerns have beed addressed. I have raised my score.

---

> > > > ### Author Response · Authors · 2025-11-25
> > > > **Thank you for reconsidering!**
> > > >
> > > > Thank you for reconsidering! You noted that your concerns have been addressed. May we ask what, if anything, you feel is necessary to have increase your confidence that our paper is acceptable? We are keen to improve the work and are happy to try to further clarify any outstanding points.
> > > >
> > > > Thank you again for your time!

---

### Official Review · Reviewer_cvbH · 2025-10-31

**Soundness:** 1
**Presentation:** 3
**Contribution:** 2
**Rating:** 4
**Confidence:** 3

**Summary:**

This paper introduces a new axis for evaluating language models (LMs): instead of only measuring whether models can solve problems, it evaluates how models assess them in a manner similar to humans. Using more than 100 two-player games, the study compares human judgments with model predictions on two quantities: fairness (expected payoff) and funness.
1) Non-reasoning LMs produce similar judgments to each other but differ from humans and gameplay-based baselines on fairness.
2) Reasoning LMs (with chain-of-thought(CoT)) show closer alignment with human judgments than non-reasoning models.
3) The alignment with human judgment is not monotonic: as models move closer to optimal play, similarity to human judgments can decrease.
4) For funness, models often reference similar factors in their reasoning traces, yet their final scores still differ.
5) Reasoning-token usage varies widely across models and shows weak association with game novelty and with distance from optimal or human predictions.

**Strengths:**

1) The game collection size, the number of human responses per game, and the number of model samples per condition are all within a reasonable range. This allows estimation of meaningful statistical signals and enables analysis of variance across games and models.
2) The evaluation covers a broad set of models, including non-reasoning and reasoning variants. The consistent observation that reasoning models show closer alignment to humans than non-reasoning models suggests that explicit reasoning traces have a measurable effect on judgment behavior.
3) Two evaluation targets, “fairness” and “funness,” which provide additional structure beyond standard goal-oriented problem solving. This creates a concrete way to compare models and humans on perceptual or judgment-driven properties, relevant to studying whether models can serve as proxies for human judgments in cognitive science settings.

**Weaknesses:**

1. Human and game selection reporting are incomplete. The main conclusions depend on the properties of both the game set and the human participants. However, the paper only states that it uses 121 games from Zhang et al. (2024a) and Collins et al. (2025) and that each game received approximately 20 human responses. There is no information about participant recruitment, demographics, incentives, exclusion criteria, or screening. Because judgments of fairness and funness are subjective, missing details about the human population make it difficult to assess how general the findings are. A similar concern applies to game selection; no justification is given for why these games are representative of broader gameplay settings.
2. Ambiguous use of the term “novel.” The text refers to the 121 games as “novel,” but they are sourced from prior work. If “novel” means new to participants or models, this should be specified. Without clarification, readers may incorrectly infer that the authors introduce a new game set.
3. Payoff scale design lacks justification. The payoff query uses two separate probability questions on a 0 – 100 scale (win given no draw, and draw) and later maps these to an expected payoff in [−1, 1]. The paper does not explain why this continuous format is preferred(especially when it will give larger variance) over a simpler ternary label {−1, 0, 1} or why no explicit uncertainty option is included.
4. Non-monotonic model-human alignment is described but not explained. The paper reports that similarity to human judgments can decrease as models approach game-theoretic optimal play. However, no mechanism is proposed. Without additional analysis, this pattern is descriptive only. It remains unclear whether the divergence stems from structural differences between human and optimal strategies, prompt effects, or sampling artifacts.
5. Positioning relative to prior LM-as-a-judge work is limited. The study contrasts non-reasoning models with models using CoT and notes that reasoning improves alignment. Similar patterns have been discussed in other LM-as-a-judge settings. The paper would benefit from situating its contribution more precisely within that literature rather than relying mainly on internal comparisons.
6. Funness construct lacks shared operational definition. Participants/models are explicitly told they may define funness however they wish. Without a shared definition or anchoring examples, different participants may use incompatible internal criteria. This increases variance in the target signal and makes it unclear what construct models are attempting to approximate. While the paper analyzes which factors appear in reasoning traces, there is no attempt to identify a stable latent structure of funness or to evaluate whether model factors correlate with participant factors.
7. Typo. There are many typos and I will just list some of them
- “evaluate the the expected payoff and funness” -> remove the duplication (Section 3.2)
- "Afer you feel like you understand the game, you can provide your response." -> After (A3.2 SYSTEM PROMPT)

**Questions:**

see Weaknesses section

**Details Of Ethics Concerns:**

This paper mentioned they have "450 participants" no information about how they recruit them and how they use this resource. Just mentioned "20 people per query".

---

> ### Author Response · Authors · 2025-11-20
> **Thank you for your review!**
>
> We thank the Reviewer for recognizing that our paper studies a “broad” set of models, games, and questions, and recognizes that our findings are “relevant to studying whether models can serve as proxies for human judgments in cognitive science settings”. We appreciate your questions, which motivated new experiments that we believe have strengthened our work. We have updated our paper directly (see the latest PDF revision) with new figures and text marked in red.
>
> However, we wanted to start by raising and clarifying a major misconception: we did not collect new human data in this work. The human data is from prior cognitive science papers (Zhang et al, 2024; Collins et al, 2025), which also include the relevant information on participant recruitment and compensation. As our submission does not involve any new human subjects research we do not require any ethics review.
>
> “A similar concern applies to game selection; no justification is given for why these games are representative of broader gameplay settings.”
> - We appreciate the concern over the choice of games studied. While the space of 121 games is a small subset of the complete space of board games or tasks, the games are nonetheless still quite varied in both mechanics and overall complexity (see response to Reviewer Tg2R as well). We don’t intend to claim that every pattern we find among these game variants would extrapolate to more complicated domains, but rather wish to highlight some of the surprising findings that emerge even in this restricted space and encourage further research into the evaluation of novel problems by LLMs.
>
> “Ambiguous use of the term “novel.” The text refers to the 121 games as “novel,” but they are sourced from prior work. If “novel” means new to participants or models, this should be specified.”
> - Good question! By “novel”, we mean new to the participants and models (i.e., that the games would not be in the models’ training data because they were explicitly generated in recent prior work. We have added a clarification to the paper. We note though that we do describe novelty, in relation to prior work as well, in our additional related work section in Appendix A1.
>
> “Payoff scale design lacks justification. The payoff query uses two separate probability questions on a 0 – 100 scale (win given no draw, and draw) and later maps these to an expected payoff in [−1, 1]. The paper does not explain why this continuous format is preferred(especially when it will give larger variance) over a simpler ternary label {−1, 0, 1} or why no explicit uncertainty option is included.”
> - As mentioned, we use the same format and human data as in the original Zhang et al 2024 and Collins et al 2025 studies. The question phrasing from their original work is cast as a probabilistic judgment/likelihood, as opposed to an instantaneous prediction of a game outcome.
> - However, during the response time, we have run and added a new prompt sensitivity analysis (see Appendix A7) wherein we both change the question phrasing and order of questions, and re-assess a subset of models’ judgements.
>
> “Non-monotonic model-human alignment is described but not explained [...] However, no mechanism is proposed.”
> - Great point! Indeed, we have not yet uncovered the mechanisms underlying this phenomenon, but we believe it is an important finding to surface this relationship in the first place. Identifying the mechanisms is out of scope for our present study; however, we do point to the similarity comparisons with other tree-search based models as one way to start to understand what these models may be doing. It is notable that the predictions of the o3 model align with the a fast and flat (Intuitive Gamer) model, and that some of the earlier (pre-reasoning) language models are perhaps more similar to judgements made by a random gameplay agent.
>
> “Positioning relative to prior LM-as-a-judge work is limited.”
> - We do include a full paragraph in the related work in Appendix A1 reviewing related LLM-as-judge work, with respect to our paper. We now better signpost to that section in the main text. Critically, our work looks at models’ evaluations of whether problems are worth solving, not LLMs’ judgements about, for instance, the solution of problems (as is typically the case for LLM-as-judge approaches).

---

> > ### Author Response · Authors · 2025-11-20
> > **Response part 2**
> >
> > (extended response due to character limit)
> > “Funness construct lacks shared operational definition. Participants/models are explicitly told they may define funness however they wish.”
> > - As described in our response as well to Reviewer Ra3f, this is partly by design! Funness is ambiguous in how it is defined (“hard to quantify”); as such, part of the work in our evaluation is to assess how models implicitly define funness – and whether that aligns to people and that of other models. We take initial steps to explore the factors models consider (in cases where we can inspect CoT or reasoning traces) in Table 2 and Appendix A6. Indeed, as you note, the original human data collection studies did not directly elicit the factors people consider; as such, we cannot compare models to peoples’ factors. However, we see further factor-level comparison as important grounds for next work – and our initial model-model comparison on factors as a valuable initial empirical comparison to get such studies off the ground!
> >
> > We have also fixed the typos you (and other reviewers’) pointed out. Thank you for reading the paper so closely!
> >
> > And we are happy to provide additional details if needed as you consider our paper further – thank you!

---

> > > ### Author Response · Authors · 2025-11-25
> > > **Following up on misunderstanding**
> > >
> > > Dear Reviewer cvbH,
> > >
> > > Thank you again for your time with our paper in the first review!
> > >
> > > We wanted to follow-up to see whether you had further questions, based on our response? We'd particular appreciate any reconsideration of the scores in your review, given the misunderstanding over whether we require ethics review (we did not collect new human data in this work).
> > >
> > > Please let us know if you have further questions -- we are happy to discuss if so!

---

> > > > ### Comment · Reviewer_cvbH · 2025-11-26
> > > >
> > > > Dear authors, thank you for your response. I also want to apologize for misunderstanding how the human data was collected. I am genuinely sorry about that.
> > > >
> > > > I have two additional comments:
> > > > 1. “We do not intend to claim that every pattern we find among these game variants would extrapolate to more complicated domains, but rather wish to highlight some of the surprising findings that emerge even in this restricted space and encourage further research into the evaluation of novel problems by LLMs.”
> > > > My concern here is about how far one can generalize from this subset of games, and what scope of tasks is appropriate for drawing broader conclusions. It would help to clarify the acceptable range of generalization and what properties of a task space should hold before conclusions from this study remain valid.
> > > >
> > > > 2. “Critically, our work looks at models’ evaluations of whether problems are worth solving, not LLMs’ judgements about, for instance, the solution of problems (as is typically the case for LLM-as-judge approaches).”
> > > > In my view, these evaluations still fall under the broader category of LLM-as-a-judge. The output format may differ, but the underlying operation is still a judgment over a structured query. There are many other judgment types that fit this template, such as whether a problem meets ethical constraints, whether a problem reflects certain considerations, or whether a problem induces more or fewer issues relative to alternatives. Your framing does provide a new angle, but the novelty of this axis is somewhat limited because it remains close to existing patterns of LLM assessment.

---

> > > > > ### Author Response · Authors · 2025-11-26
> > > > > **Follow-up to questions and continued misconception**
> > > > >
> > > > > Dear Reviewer cvbH, we appreciate you acknowledging part of the misunderstanding! May we ask that you please revise your score to note that we do NOT need ethics review? As noted, we did *not* collect new human data and therefore do not warrant ethics review.
> > > > >
> > > > > We also appreciate your additional questions/comments. We respond to each in turn:
> > > > >
> > > > > “My concern here is about how far one can generalize from this subset of games, and what scope of tasks is appropriate for drawing broader conclusions. It would help to clarify the acceptable range of generalization and what properties of a task space should hold before conclusions from this study remain valid.”
> > > > > - Good point! We do acknowledge, as noted in our response and our Discussion, that the games we consider are limited. However, they do cover a range of interesting two player strategic games. We expect that our results likely hold for competitive tasks completed over relatively short-time scales that permit goal-directed heuristics. While this class is somewhat narrow, it extends beyond just the specific variants studied in this paper (e.g., potentially to games like Hex or Othello). Yet, this is an empirically testable question! It is less clear, however, if the results will generalize to other non-competitive strategic contexts (e.g., involving cooperation),contexts where players do not know if they are engaging with competitors or collaborators, or if games that involve more than two players. These are all important areas of study! We hope our work is one initial stepping stone that motivates such further work, expanding and probing generalization against other games and game categories – and informing what new kinds of human data researchers in cognitive science may collect as well. Perhaps most immediately, it is an interesting question whether the results generalize to other two-player competitive board games. We have added a clarification on these limitations and open questions to our discussion (see the red text in the updated PDF).
> > > > >
> > > > > “In my view, these evaluations still fall under the broader category of LLM-as-a-judge.”
> > > > > - We hear your point on this. We do think – as you said that our “framing does provide a new angle.” Indeed, we are effectively treating the LLM as a “judge” of the game evaluations, but arguably offering a perspective (see discussion with Reviewer AE1w) on the kinds of evaluations we may probe LLMs on. We include a section on LLM-as-judge work in our Appendix Related Work section, though we are open to moving this to the main text if you think necessary for our work to be appropriately contextualized.
> > > > >
> > > > > Please let us know if you have further questions!

---

### Official Review · Reviewer_AE1w · 2025-11-01

**Soundness:** 2
**Presentation:** 3
**Contribution:** 2
**Rating:** 4
**Confidence:** 5

**Summary:**

The authors explore and evaluate how frontier language models evaluate games. The bigger picture is that, as a field, we demonstrate that models can play and learn in games, but can they also choose which game to play, and if so, based on what? Do their tastes in games match human preference, or do they qualitatively differ? The paper collects human annotations for game evaluation for two specific questions. The frontier models are asked the same questions. The paper analyses the (dis)agreement between humans and frontier models, as well as the intra-frontier model disagreement. The main contribution is probably the extensive statistical analysis of human and model preference. The most interesting takeaway seems that reasoning in models leads to stronger correlation with human notions of fun, at least for these specific games, in a specific annotation setting.

**Strengths:**

## Clarity
- The paper presents its results transparently and thoroughly.
- The paper is clear about hyperparameters and which prompts were used.
- The authors are transparent with the limitations.
- The paper is mostly easy to follow.
- The figures are easy to understand and mostly well formatted.
- The authors compare and contrast with relevant related work.

## Quality
- The statistical analysis is thorough, and meaningful measures of variance are provided. The details of how they were computed were also documented.
- The experiments are thorough, cover a lot of frontier models, and investigate the role of reasoning (both through CoT and trained reasoning models).
- The evaluation goes the extra mile and also looks at token usage, which was interesting!

## Originality and Significance
- The broader topic is definitely interesting and goes into capturing human subjectivity and open-ended agentic problems. The actual study setup can also be interesting to game developers.

**Weaknesses:**

## Clarity
1. The reviewer struggles to understand the motivation for the two dimensions of evaluations as outlined in the Intro. It's unclear how the second paragraph "highlights" (line 48) two dimensions of evaluations. For example, Hunicke (2004) appears to discuss the MDA framework, rather than the two dimensions of game evaluation. Isn't something hard to quantify necessarily hard to compute? What is an example of something hard to quantify but easy to compute? For example, in Figure 1, the example question is "What kind of game is this?" While being more of a classification task than a quantification task, why is that easier to compute? Currently, it's the reviewer's opinion that "Hard to compute" is too hard to quantify and "Hard to quantify" is too hard to compute to provide a useful framework for game evaluation by models.
2. For the question about expected payoff, it was not clear from the main text that the prompt that humans and models were asked was:

> Then, for each game, your task is to answer: assuming both players play *reasonably*

The Introduction describes the question as if it's focused *only* on the hard-to-compute part, i.e., all humans and models have to do is attempt to compute the game-theoretically optimal outcome for two rational players.

> For each game, we test a series of language and reasoning
models using two reasoning queries that engage both dimensions of evaluation: one that is difficult
to compute and another that is difficult to compute and quantify.

and also in Section 3.2:

> We prompted a series of language and reasoning models to evaluate the the expected payoff and funness of each of the 121 games


However, the question asks explicitly for *reasonable*, which, at least to the best of the reviewer's understanding, is not a well-defined term. For example, when playing against a child, where you might let them win on purpose, reasonable gameplay looks different from rational gameplay. In the reviewer's opinion, the expected payoff question then also becomes a question that is both hard to quantify *and* hard to compute. This muddies the comparison to the game-theoretic optimal performance. The results do seem to suggest that humans and frontier models tend to infer "rational", but for example, the reasoning trace from DeepSeek on line 1538 explicitly considers the agents to be reasonable, not rational.

3.  For the annotation study, it's unclear in what order the human annotators were provided the questions. Would the same humans always answer both question 1 and question 2, and in the same order? If so, answering question 1 might bias a human to think about payoffs in the funness estimates. Ideally, annotators would be asked either question 1 or question 2 about a game to avoid a bias that wouldn't be present in the models' context.

## Originality and Significance
1. In the Conclusion, the authors write

> We laid out a framework for thinking about evaluation of AI systems’ capacity for problem
evaluation

and the Abstract states:
> In this paper, we advocate for a new paradigm that
assesses AI systems’ evaluation of games. First, we introduce a formalism for
evaluating such evaluations. We then leverage a large-scale dataset of over 100 novel board games and over 450 human judgments


However, it's unclear to the reviewer what exactly the framework or new paradigm is. Section 2 explains that games may have factors that you can use to judge a game by. However, there is nothing specific to games in the framework described in Section 2. Anything that might be evaluated may have multiple factors (dimensions) by which to judge it. Is the framework the two specific questions, i.e., should an agent always evaluate expected payoff and fun? Is it the two dimensions? At the moment, to the best of the reviewer's understanding, the framework is simply gathering human annotations on a subjective measure, asking LLMs the same question and measuring correlation for a very specific domain. There is no evidence that any of these insights would generalise or that these dimensions (or questions) prove useful in "evaluating language models' evaluations of games".

To summarise, the paper's correlation analysis is well-executed; however, it's challenging to see how this paper contributes anything more general than a correlation analysis with human notions of fun for a specific family of games.

**Questions:**

1. Figure 2 caption says "modle-predicted"
2. There's a typo in the example prompt, i.e., see line 981 "For each quetsion, provide your a single number"
3. The y-axes ticks in Figure 4b are very small.

The reviewer has no problem with the technical execution of the paper. If the authors could address the concerns raised in Weakness, the reviewer would be very grateful and would look forward to the rebuttal period.

---

> ### Author Response · Authors · 2025-11-20
> **Thank you for your review! (part 1)**
>
> We thank the Reviewer for recognizing the technical merit of our paper (“The reviewer has no problem with the technical execution of the paper.”; “The paper presents its results transparently and thoroughly.”) and that “the broader topic is definitely interesting.”
>
> We particularly appreciate the thoroughness of the Reviewer’s comments and questions! Your points motivated new experiments that we believe have strengthened our work. We have updated our paper directly (see the latest PDF revision) with new figures and text marked in red. We respond to each of your points in turn, focusing on the two key topics you raised.
>
> Prompt sensitivity and question phrasing:
>
> “Would the same humans always answer both question 1 and question 2, and in the same order?”
> - Good question! Participants in the original human study always saw question 1 then 2. To begin to assess the potential differences in order, we ran o3 (the most human-aligned model) for 20 rollouts on the same 121 games, with the order of the payoff questions flipped. We find the final payoff is highly correlated with the payoff of the original order. We include this analysis in Appendix A7.2. Of course, people may respond differently – and that’s a worthwhile question to assess! But it is beyond the scope of our present study.
>
> “However, the question asks explicitly for reasonable, which, at least to the best of the reviewer's understanding, is not a well-defined term.”
> - Excellent point! We followed the prompt instructions of the original human study, which asked people to assume reasonable play. However, you are right that this could influence model predictions and is somewhat unclear. Inspired by your comments, we ran a new prompt sensitivity analysis (now in Section A7.1), where we replaced the text to assume “reasonable” play with “optimal” play. While we observe little difference in judgements for o1 and GPT-5 relative to the game-theoretic optimal, we find a marked difference for o3’s predictions. This does suggest, as you were raising, that there is ambiguity in how one (a person, or model) interprets the question and that interpretation may also not be aligned, with perhaps o3 being more adept at simulating real human judgements (when tasked to imagine “reasonable” players) compared to GPT-5 where “reasonable” seems similar, in its interpretation, to “optimal” (at least at the surface level of judgement correlation). Moreover, all models fit human judgements worse when judging under the assumption of optimal over reasonable play, suggesting that the models may be reasoning about such questions differently depending on the kind of player behavior in question. We have added additional clarification in our main text and Appendix on these points and are excited about future work inspired by our study to better characterize models’ perceptions of player behavior.

---

> > ### Author Response · Authors · 2025-11-20
> > **Further response (part 2)**
> >
> > (extending due to character limit!)
> >
> > Framework and contributions:
> >
> > “ it's unclear to the reviewer what exactly the framework or new paradigm is [...]  Is the framework the two specific questions, i.e., should an agent always evaluate expected payoff and fun? Is it the two dimensions? [...] There is no evidence that any of these insights would generalise or that these dimensions (or questions) prove useful in "evaluating language models' evaluations of games".”
> > - Based on your feedback, we recognize the language that we propose a new “framework” is not quite right. We have changed “framework” to “perspective”. A key contribution of our work is a new lens (as pointed out by Reviewer Tg2R) for thinking about evaluation of models’ reasoning: one which focuses not just on problem solving but on evaluating whether a problem is worth solving. We provide some notational backing in Section 2 to define this problem which we viewed as a “conceptual” framework when coupled with our categorization of two dimensions (see below). The framework is not the gathering of the human data per say, but the concept of comparing human and models’ evaluations of the games themselves. And then we conduct an initial empirical investigation of model judgements under this perspective.
> > - While we do instantiate this evaluation first in the context of a relatively constrained set of initial games, as we note in our responses to Reviewers Tg2R, the games we consider are quite rich and varied within this set. We would argue that we already are getting insight into “evaluating language models' evaluations of games” with our present evaluations. We find that many of the latest reasoning models are more human-aligned, and more aligned to other tree search-based models; yet, becoming more aligned to people can also come with tradeoffs (e.g., being less aligned to the game-theoretic predictions), raising critical questions about the effects of optimizing models towards either “perfect rationality” or human-like behavior.
> > - Moreover, the specific questions we look at – payoff / fairness and fun – are two of the most critical people have asked about games throughout history and other fields that look at games, e.g, in game theory. We think assessing models’ evaluations of these questions is therefore highly valuable and a natural set of questions to look at when first looking at models’ evaluations of whether a game is valuable to play. However, we are not suggesting that people nor models should always assess particularly those two questions, but rather, encourage more research around the question of asking questions before engaging in a problem (like people do!).

---

> > > ### Author Response · Authors · 2025-11-20
> > > **Further response (part 3)**
> > >
> > > “Currently, it's the reviewer's opinion that "Hard to compute" is too hard to quantify and "Hard to quantify" is too hard to compute to provide a useful framework for game evaluation by models.”
> > > - We appreciate you raising confusion on our dimensions. We have revised the text in the paper (see revised PDF) to try to better clarify the dimensions. The goal of these dimensions is to outline that different queries may be more or less interesting (and more or less easy) to evaluate model and human judgements on.
> > > “Isn't something hard to quantify necessarily hard to compute?”
> > > - It’s possible for something to be hard to quantify (e.g., to determine the right measures to assess) yet easy to compute each of those measures. For instance, utilitarian ethics may have this quality. It might be hard to quantify what values to assign to different quantities (such as human lives or irreplaceable works of art), but it is easy to compute the sum of these values when considering a possible outcome of an action.
> > > “For example, Hunicke (2004) appears to discuss the MDA framework, rather than the two dimensions of game evaluation.”
> > > - Thank you for raising this. As we noted in our response to Reviewer Tg2R as well,  the cite is not well-placed (rather than the introduction of the dimensions). We did not intend for our two-dimension format to directly tie to MDA-constructs. Rather, we were citing that paper for its discussion of the ambiguity of funness.
> > > “it's challenging to see how this paper contributes anything more general than a correlation analysis with human notions of fun for a specific family of games”
> > > - Our analyses include but go beyond notions of just correlational analyses with human notions of funness. As discussed above, a key contribution is laying out an underappreciated perspective that understanding AI systems’ reasoning capacities should assess not just problem solving (gameplay) but whether games are worth playing at all. Arguably, our payoff prediction analyses – and finding that there is a non-monotonic relationship between several models and the closeness of their judgements to people and game-theoretic optimal values – is a more valuable contribution than our funness correlations. Moreover, we point out important and peculiar open questions into the relationship between compute usage and judgements. Overall, we hope more researchers, inspired by our work, further study evaluations of models’ evaluations of problems, games, and beyond; we see our work as laying out a broad research program that we and others can and should do more work in.
> > >
> > > We have also fixed the typos you (and other reviewers’) pointed out. Thank you for reading the paper so closely!
> > >
> > > And we are happy to provide additional details if needed as you consider our paper further – thank you!

---

> > > > ### Comment · Reviewer_AE1w · 2025-11-24
> > > >
> > > > The additional experiments are greatly appreciated. The reviewer agrees with the authors that the additional results and clarification strengthen the paper substantially.
> > > >
> > > > Regarding the text, for example, in the intro, would it be possible to clarify the sentence
> > > > "one that is difficult to compute and another that is difficult to compute and quantify." to highlight further that the "reasonable" play is also somewhat hard to quantify. Given the empirical results, it's probably fine to leave as is for the story. Still, the reviewer would appreciate a short acknowledgement of the "reasonable" phrasing in the main body, not just in the appendix.
> > > >
> > > > After that, the reviewer is more than keen to update the score to reflect the improvements.

---

> > > > > ### Author Response · Authors · 2025-11-24
> > > > > **Updated main text**
> > > > >
> > > > > Thank you for your consideration! We have updated that section of the introduction. We appreciate the time you've taken with our paper!

---

> > > > > > ### Comment · Reviewer_AE1w · 2025-11-24
> > > > > >
> > > > > > The reviewer adapted the rating, soundness, presentation, and contribution scores accordingly.
> > > > > >
> > > > > > The reviewer is yet pondering the hard-to-quantify axis and might come back to it.

---

> > > > > > > ### Author Response · Authors · 2025-11-24
> > > > > > > **Thank you!**
> > > > > > >
> > > > > > > Thank you so much for reconsidering your score! We are also happy to engage further on the axes (we appreciate the helpful comments and questions).

---

### Official Review · Reviewer_Tg2R · 2025-11-02

**Soundness:** 2
**Presentation:** 3
**Contribution:** 3
**Rating:** 4
**Confidence:** 4

**Summary:**

The paper propose to evaluate whether models can evaluate games (as benchmarks) themselves. Using 121Tic-Tac-Toe variants of board games and ~450 human judgments, the authors compare non-reasoning LMs, reasoning LMs (with traces), and simulation agents, and report a non-monotonic trade-off: models closer to game-theoretic optimality, their capacity as (game) evaluation judges align less with human judges. They also observe higher variance for funness evaluation and highlight the need for resource-rational meta-reasoning.

**Strengths:**

1. Evaluating the games (as evaluation benchmarks) themselves (for design properties like payoff, funness) is a novel and valuable lens.

2. Includes non-reasoning LMs, reasoning models, and rule-based (MCTS) agents, contextualizing model judgments against humans and game-theoretic references.

3. Systematically contrasts AI vs human evaluations, revealing the non-monotonic alignment for game-theoretic optimal LLM players.

**Weaknesses:**

1. There lacks evaluation metrics beyond R^2. Would be useful and informative to add Spearman’s rho and Kendall’s tau to capture rank alignment.​

2. There lacks a strong justification on why choosing MDA-style constructs for formalizing "funness". Could the authors provide more discussion on their choice?

3. The 121 games are largely tic-tac-toe variants.

4. Lacks a prompt sensitivity study and analysis.

**Questions:**

1. Would make the paper stronger if the authors could report preliminary results for expert-refined vs auto-tuned prompts (e.g. with DSPy), since prompting is a major driver of judgments.

2. Could the authors include other rank-based metrics (Spearman/Kendall) or motivate why it suffices to sticking with R^2 as evaluation metrics?

3. Does the human-vs-optimal non-monotonicity persist on cooperative or asymmetric games? Any early evidence?
arXiv

4. Do the authors have comments on how do these results transfer to other evaluation settings (beyond game-as-an-eval, for other benchmark/test construction)?

---

> ### Author Response · Authors · 2025-11-20
> **Thank you for your review!**
>
> We thank the Reviewer for recognizing that our paper offers a “novel and valuable lens” which “systematically contrasts AI vs human evaluations.” We appreciate your questions, which motivated new experiments that we believe strengthen our work. We have updated our paper directly (see the latest PDF revision) with new figures and text marked in red. We respond to each of your points in turn and want to clarify the misconception that we only consider R^2 (we consider several other measures, in the main text and Appendix).
>
> “There lacks evaluation metrics beyond R^2. Would be useful and informative to add Spearman’s rho.”
> - Based on your suggestion, we computed and included Spearman rho calculations for the payoff and funness ratings (see Appendix A4). We focused on R^2 in the main text as it has a clearer semantic interpretation (i.e., amount of explainable variance), and R^2 is a widely considered good, standard measure for assessing model-human fit in computational cognitive science.
> - However, we also want to point out we do include other substantial analyses that do not focus only on R^2. Table 1 also reports accuracy and the mean absolute difference between the predicted and game-theoretic optimal payoff. Sections A4 and A5 of our Appendix also include several additional analyses into predicted judgements and reasoning token usage that do not focus exclusively on R^2.
>
> “There lacks a strong justification on why choosing MDA-style constructs for formalizing "funness".”
> - Thank you for raising this. We did not intend for our two-dimension format to directly tie to MDA-constructs. Rather, we were citing that paper for its discussion of the ambiguity of funness. We think the cite is not well-placed and removed it from the introduction.
>
> “Does the human-vs-optimal non-monotonicity persist on cooperative or asymmetric games?”
> - Good question! To our knowledge, we are the first to surface this non-monotonic behavior in the relationship b/w AI systems’ and people’s judgements compared to AI systems and a notion of optimality. This finding (and the lens to do the evaluations that can help surface such findings) is a key contribution of our work. We are excited by future work, inspired by our work, which can design new games (and collect appropriate human data) to further probe the relationship between AI judgements, people, and notions of optimality in the future.
>
> “The 121 games are largely tic-tac-toe variants.”
> - This is indeed a limitation, which we acknowledge in Section 6. We hope more researchers expand the classes of games, and kinds of game queries, when evaluating not just play but the evaluations models make of games before any play!
> However, even though the games can be cast as variants of tic-tac-toe, they do span quite a wide range of mechanics that lead to rich strategic behavior (see Table A2). Even though we focus on a single class of two-player competitive strategy games here, we do consider many games within that class – in contrast to many works in AI and psychology which focus on just one game. We see value in having a somewhat constrained yet still rich testbed, where already there are substantial insights to be gleaned into the ways models (and people) evaluate these games before any play! And despite being variants of tic-tac-toe, the games capture many hallmarks of real-world decision making and planning problems.
>
> “Lacks a prompt sensitivity study and analysis.”
> - Based on your + other Reviewers’ feedback, we ran a series of prompt sensitivity analyses. We considered prompt variants where we (1) replace the instruction to imagine “reasonable” players with “optimal” play, and (2) flip the order of the payoff questions.  See Appendix A7 for analysis.
>
> “Do the authors have comments on how do these results transfer to other evaluation settings (beyond game-as-an-eval, for other benchmark/test construction)?”
> - Great question! If we think of games as microcosms of the many systems of rules and reward that people engage with in life, then these kinds of evaluations (e.g., of the fairness or funness) of rule structures may have implications when understanding how models reason about whether to engage with or even change new rules. This could be relevant in AI-based mechanism design (e.g., Chris Summerfield and colleagues’ work) or also for the assessment of whether problems are worth engaging with (e.g., in mathematics). A key contribution of our paper is even raising awareness of this distinction b/w evaluating problem solving and evaluating whether a problem is worth solving. We need more evaluations of the latter, and our work is a first step in this direction. We also expect results to transfer to evaluating language models as helpful assistants: for tasks like planning travels and suggesting activities, knowing what is fun and fair for humans is essential for alignment.
>
> Thank you for further considering our paper. We are happy to expand on any of these points further, if helpful!

---

> > ### Author Response · Authors · 2025-11-25
> > **Response consideration?**
> >
> > Dear Reviewer Tg2R,
> >
> > Thank you again for your thoughtful comments and questions in your initial review! We wanted to check whether the additional experiments and clarifications in our response alleviate your concerns? If not, we are happy to try to clarify further and/or potentially explore additional analyses to make our paper stronger.
> >
> > Thank you!

---

### Author Response · Authors · 2025-12-04
**Summarizing response period**

Dear Reviewers and ACs,

We thank you all for your time, amidst a challenging situation! We wanted to summarize the primary feedback and the substantial changes we made to our paper as part of the response period.

We have uploaded a **revised PDF** to OpenReview, with changes made during revisions **marked in red**.

We thank the reviewers for recognizing:
- The **novel perspective** our framework brings to understanding LM reasoning and its relation to human reasoning (Tg2R, cvbH, Ra3f)
- The **comprehensiveness** of our experiments (all reviewers – Tg2R, AE1w, cvbH, Ra3f)
- And generally having chosen an **interesting** topic to study (Tg2R, AE1w)

In their initial reviews, reviewers’ comments focused on a **few main areas**:
- Prompt sensitivity
- Clarification and value of the conceptual framework
- Inquiries about additional empirical analyses (e.g., into the relationship between reasoning amount and game-level predictions)
- Variety of games and generality of results across game categories

**Prompt sensitivity**

To address these concerns, we **ran several new experiments, varying model prompts** (see responses to Reviewers Tg2R, AE1w, and Ra3f in particular). See new Appendix Section A7.

**Clarification and value of the conceptual framework**

We also **revised our description of the two dimensions** we use to carve the space of game evaluation queries in the main text of our paper. We had a lively discussion with Reviewer AE1w, leading them to **raise their score from a 4 to an 8**!

**Additional empirical analyses**

We also **expanded the cross-model game analyses** in the Appendix (see responses to Tg2R and Ra3f). Ra3f **raised their score to a 6** in response. We didn’t hear from Reviewer Tg2R before the incident, but are hopeful they would have responded similarly positively to our new experiments.

**Variety of games**

While our initial responses better highlighted the **diversity in the 121 games we focus on**, we excitingly have just finished running a **new set of experiments expanding the game classes** we consider. We ran a subset of the LMs on 16 variants across four new game categories (variants of Reversi, Hex, Yavalath, and Cooperative Tic-Tac-Toe, respectively). The first three games are new kinds of competitive games and the latter a cooperative game (to address Ra3f in particular). Given the time it took to develop and run experiments on new game variants, unfortunately we weren’t able to share these results with the reviewers during the pre-incident period; however, **we are hopeful that the reviewers would have enjoyed these new results!** We have included them in Appendix Section A8. The differences between non-reasoning and reasoning LMs do generalize to cooperative variants, but are more mixed for the other competitive variants we consider. **These examples demonstrate that our novel evaluation framework is extendable and generalizable.** More investigation is needed into these games, and we hope our work inspires further evaluation of not just game play, but game evaluation into these and other game categories.

**Miscellany**

We also have now **added additional clarification on the MCTS runs** (see Appendix Section A4.1) following questions from Ra3f. We did not have these during our active discussion, but are hopeful it could have increased Ra3f’s confidence in our paper more.

We also wanted to highlight that Reviewer cvbH had several **misconceptions** about our work. We did not run any new human experiments, and therefore do not require ethics review, as they marked. We also continued to clarify the positioning of our paper relative to prior work, but unfortunately our discussion was cut short due to the incident. We are hopeful Reviewer cvbH would be happy with the changes we made since.

Thank you for your time evaluating our work!

---

### Meta-Review · Area_Chair_9Dt2 · 2026-01-07

**Summary:**

This paper proposes a novel paradigm to assess AI’s ability to evaluate games (payoff/fairness and funness) rather than just play them, using 121 Tic-Tac-Toe variants and 450+ human judgments. Key findings include reasoning models’ better human alignment, non-monotonic alignment with game-theoretic optimality, and unpredictable resource usage by reasoning models. Reviewers recognized the novel meta-reasoning perspective and comprehensive experiments but raised critical concerns about game diversity, prompt sensitivity, framework clarity, and human data transparency. The rebuttal effectively addressed most concerns. Overall, the paper presents an interesting perspective on evaluating LLMs for meta-reasoning, and the strengths of the work now outweigh the remaining gaps, justifying acceptance.

Besides, this is a very active research area. There have been quite a few recent works before this submission. To name a few:

Lin et al, GAMEBoT: Transparent Assessment of LLM Reasoning in Games, ACL 2025

Duan et al, GTBench: Uncovering the Strategic Reasoning Limitations of LLMs via Game-Theoretic Evaluations, NeurIPS 2024.

Wu et al, SmartPlay: A Benchmark for LLMs as Intelligent Agents, ICLR 2024

Huang et al, How Far Are We on the Decision-Making of LLMs? Evaluating LLMs' Gaming Ability in Multi-Agent Environments, ArXiv preprint arXiv:2403.11807. 2024

A more careful literature review is needed with necessary discussion on the relevance and differences with those works.

**Reviewer Concerns:**

Key Concerns Raised:
- Game diversity (all reviewers): Over-reliance on Tic-Tac-Toe variants; lack of cooperative/asymmetric/non-board games undermines "general problem evaluation" claims.
- Prompt sensitivity (Tg2R, AE1w, Ra3f): No initial analysis of prompt variations (e.g., "reasonable" vs. "optimal" play) or temperature impact; limited rollouts (20 per model) raise robustness questions.
- Framework clarity (AE1w): Ambiguity in "hard to compute" vs. "hard to quantify" dimensions; "framework" mislabeled as novel with no game-specific utility.
- Human data/funness operationalization (cvbH, Ra3f): Missing participant demographics/recruitment details; funness lacks shared definition, leading to unstructured judgments.
- Game-theoretic validity (Ra3f): Only 78/121 games had computable optimal values; no MCTS convergence criteria or accuracy validation.
- Prior work positioning (cvbH): Insufficient distinction from "LLM-as-a-judge" literature; novelty overstated.

Addressed in Rebuttal:
- Expanded to 16 new game variants (4 categories) and analyzed generalization; clarified mechanical diversity of existing Tic-Tac-Toe variants (Tg2R, Ra3f).
- Conducted prompt sensitivity analyses (reasonable/optimal play, question order) and added reasoning amount tests; disclosed default temperature settings (0.7 for most models, 1.0 for o1/o3/GPT-5) (Tg2R, AE1w, Ra3f).
- Revised "framework" to "perspective"; clarified dimensions with examples (e.g., utilitarian ethics as "hard to quantify but easy to compute"); removed misaligned MDA framework citation (AE1w).
- Clarified human data is from prior work (no new collection, resolving ethics concerns); analyzed models’ funness-related reasoning factors (balance, challenge, etc.) (cvbH, Ra3f).
- Provided MCTS convergence criteria (50 simulations, 1000 iterations per game) and mathematical derivation details for optimal values (Ra3f).
- Expanded related work to distinguish from "LLM-as-a-judge" (focus on problem evaluation vs. solution judgment); added main-text signposting to appendix details (cvbH).

Outstanding/Partially Addressed:
- Main experiments still rely on Tic-Tac-Toe variants; new 16 variants lack human data to validate alignment (Tg2R, Ra3f).
- Funness remains unstandardized (inherent to the query’s "hard to quantify" nature but limits interpretability of human-model alignment) (cvbH, Ra3f).
- Framework dimensions lack rigorous operationalization for broader evaluation tasks beyond games (AE1w).
- Generalization to non-competitive/non-board games unproven; no evidence of transfer to real-world problem evaluation (cvbH, Ra3f).
- Limited exploration of temperature variability and rollout sufficiency; robustness to prompt engineering untested (Ra3f).
- Non-monotonic alignment mechanism remains unexplained (cvbH).

**Reviewer Scores:**

Tg2R: 4 → likely 4 or 6.

AE1w: 4 → likely 6 or 8.

cvbH: 4 → likely 4 or 6.

Ra3f: 4 → likely 6.

---

### Decision · Program_Chairs · 2026-01-26

Accept (Poster)